# Deciphering molecular heterogeneity and dynamics of human hippocampal neural stem cells at different ages and injury states

Junjun Yao[1,2†], Shaoxing Dai[1,2†], Ran Zhu[1,2†], Ju Tan[3†], Qiancheng Zhao[1,2], Yu Yin[1,2], Jiansen Sun[4], Xuewei Du[1,2], Longjiao Ge[1,2], Jianhua Xu[3], Chunli Hou[3], Nan Li[1,2], Jun Li[2], Weizhi Ji[1,2], Chuhong Zhu[3]*, Runrui Zhang[1,2]*, Tianqing Li[1,2]*

[1]State Key Laboratory of Primate Biomedical Research, Institute of Primate Translational Medicine, Kunming University of Science and Technology, Kunming, China; [2]Yunnan Key Laboratory of Primate Biomedical Research, Kunming, China; [3]Department of Anatomy, National and Regional Engineering Laboratory of Tissue Engineering, State Key Laboratory of Trauma, Burn and Combined Injury, Key Lab for Biomechanics and Tissue Engineering of Chongqing, Third Military Medical University, Chongqing, China; [4]Zhong-Zhi- Yi-Gu Research Institute, Chongqing, China

*For correspondence:
zhuch99@tmmu.edu.cn (CZ);
zhangrr@lpbr.cn (RZ);
litq@lpbr.cn (TL)

[†]These authors contributed equally to this work

Competing interest: The authors declare that no competing interests exist.

**Abstract** While accumulated publications support the existence of neurogenesis in the adult human hippocampus, the homeostasis and developmental potentials of neural stem cells (NSCs) under different contexts remain unclear. Based on our generated single-nucleus atlas of the human hippocampus across neonatal, adult, aging, and injury, we dissected the molecular heterogeneity and transcriptional dynamics of human hippocampal NSCs under different contexts. We further identified new specific neurogenic lineage markers that overcome the lack of specificity found in some well-known markers. Based on developmental trajectory and molecular signatures, we found that a subset of NSCs exhibit quiescent properties after birth, and most NSCs become deep quiescence during aging. Furthermore, certain deep quiescent NSCs are reactivated following stroke injury. Together, our findings provide valuable insights into the development, aging, and reactivation of the human hippocampal NSCs, and help to explain why adult hippocampal neurogenesis is infrequently observed in humans.

## eLife assessment

Using state-of-the-art single-nucleus RNA sequencing, Yao et al. investigate the transcriptomic features of neural stem cells (NSCs) in the human hippocampus to address how they vary across different age groups and stroke conditions. The authors report alterations in NSC subtype proportions and gene expression profiles after stroke. Although the study is **valuable** and the analysis is comprehensive, the significance is restricted by well-acknowledged technical limitations leading to **incomplete** evidence supporting some main conclusions.

## Introduction

Continuous learning and memory formation throughout life is driven by developmental and adult neurogenesis. The dentate gyrus (DG), a part of the hippocampus and one of the main neurogenic

niches, sustains neurogenesis through the activity of resident neural stem cells (NSCs) (*Berg et al., 2019*). Although adult neurogenesis in rodents (*Hochgerner et al., 2018*; *Dulken et al., 2017*) is well studied, and age-related neurogenesis decline is conserved across species, whether hippocampal neurogenesis persists in the adult human brain has been debated over the years. Finding a conclusive answer to this question is not trivial, as available human brain tissue is rare, and analysis is fraught with technical challenges. Based on marker immunostaining, a few studies found no evidence of neurogenesis in human after adolescence (*Cipriani et al., 2018*; *Sorrells et al., 2018*; *Franjic et al., 2022*), while others detected that human neurogenesis persists in adulthood but declines during aging (*Boldrini et al., 2018*; *Moreno-Jiménez et al., 2019*; *Tobin et al., 2019*; *Terreros-Roncal et al., 2021*). It is expected that single-cell RNA-seq will help resolve the ongoing debate, as this technology is capable of bypassing the biases associated with traditional methods of immunostaining and quantification. Single-cell analysis approaches can also help identify novel cell markers and resolve the dynamics of transcriptional signatures during neurogenesis under different conditions. Leveraging these advantages, several groups performed single-nucleus RNA-seq (snRNA-seq) (*Habib et al., 2016*) analysis to investigate adult hippocampal neurogenesis in the human brain (*Franjic et al., 2022*; *Zhou et al., 2022*; *Wang et al., 2022*). Although one study failed to detect evidence of adult neurogenic trajectories in human hippocampal tissues (*Franjic et al., 2022*), the other two reported the presence of molecular programs consistent with the capacity for the adult human DG to generate new granule cells (GCs) (*Zhou et al., 2022*; *Wang et al., 2022*).

Accumulated publications support the existence of neurogenesis in the adult human hippocampus, but the homeostasis and developmental potentials of NSCs under different contexts remain unclear. Particularly, while actively proliferating in early development, mouse NSCs gradually acquire quiescent properties and transform into quiescent NSCs (qNSCs) with age. Although neurogenesis declines in the mouse aging hippocampus as a consequence of NSC loss and dormancy, qNSCs can be reactivated into active NSCs (aNSCs) that give rise to GCs which integrate into existing neural circuits (*Encinas et al., 2011*; *Obernier and Alvarez-Buylla, 2019*). Specifically, ischemic insult in the adult mouse brain has been reported to evoke qNSC to transition into an active state. However, whether these similar mechanisms occur in the human hippocampus is unknown (*Llorens-Bobadilla et al., 2015*).

To gain insight into why adult hippocampal neurogenesis is challenging to observe in humans, we believe that examining NSCs under varying conditions may be helpful, as they are the sources of neurogenesis. Thus, we conducted snRNA-seq analysis on human hippocampal tissue and investigated the heterogeneity and molecular dynamics of hippocampal NSCs across neonatal, adult, aging, and stroke-induced injury conditions. Based on comparative analysis of cell types, and developmental trajectories and molecular features of NSCs under different contexts, we found that NSCs, including qNSCs, primed NSCs (pNSCs), and aNSCs, exhibit different molecular features and dynamics across neonatal, adult, aging, and stroke-induced injury conditions. We observed a subset of NSCs that display quiescent properties after birth, and most NSCs become deep quiescence during aging. Notably, some deep qNSCs can be reactivated to give rise to pNSCs and aNSCs in the stroke-injured adult human hippocampus. In addition, we also found that immature GC markers widely used in mice studies, including DCX and PROX1, are non-specifically expressed in human hippocampal GABAergic interneurons (GABA-INs). We further identified neuroblast (NB)-specific genes *CALM3, NEUROD2, NRN1*, and *NRGN* with low/absent expression in human GABA-INs. Together, our findings provide an important resource to understand the development, aging, and activation of human postnatal hippocampal NSCs.

## Results

### Single-nucleus atlas of the human hippocampus across ages and injury

To generate a comprehensive cell atlas of neurogenic lineages in the human hippocampus, we collected 10 donated post-mortem hippocampal tissues. We then dissociated the anterior-mid hippocampus (which has an obvious DG structure) and performed 10x Genomics snRNA-seq. We also performed immunostaining for the counterpart side of each hippocampus sample (*Figure 1A*). The 10 individual samples, divided into four groups according to age and brain health, included neonatal (day 4 after birth, D4, n=1), adult (31, 32 years of age, n=2), aging (from 50 to 68 years of age, n=6), and

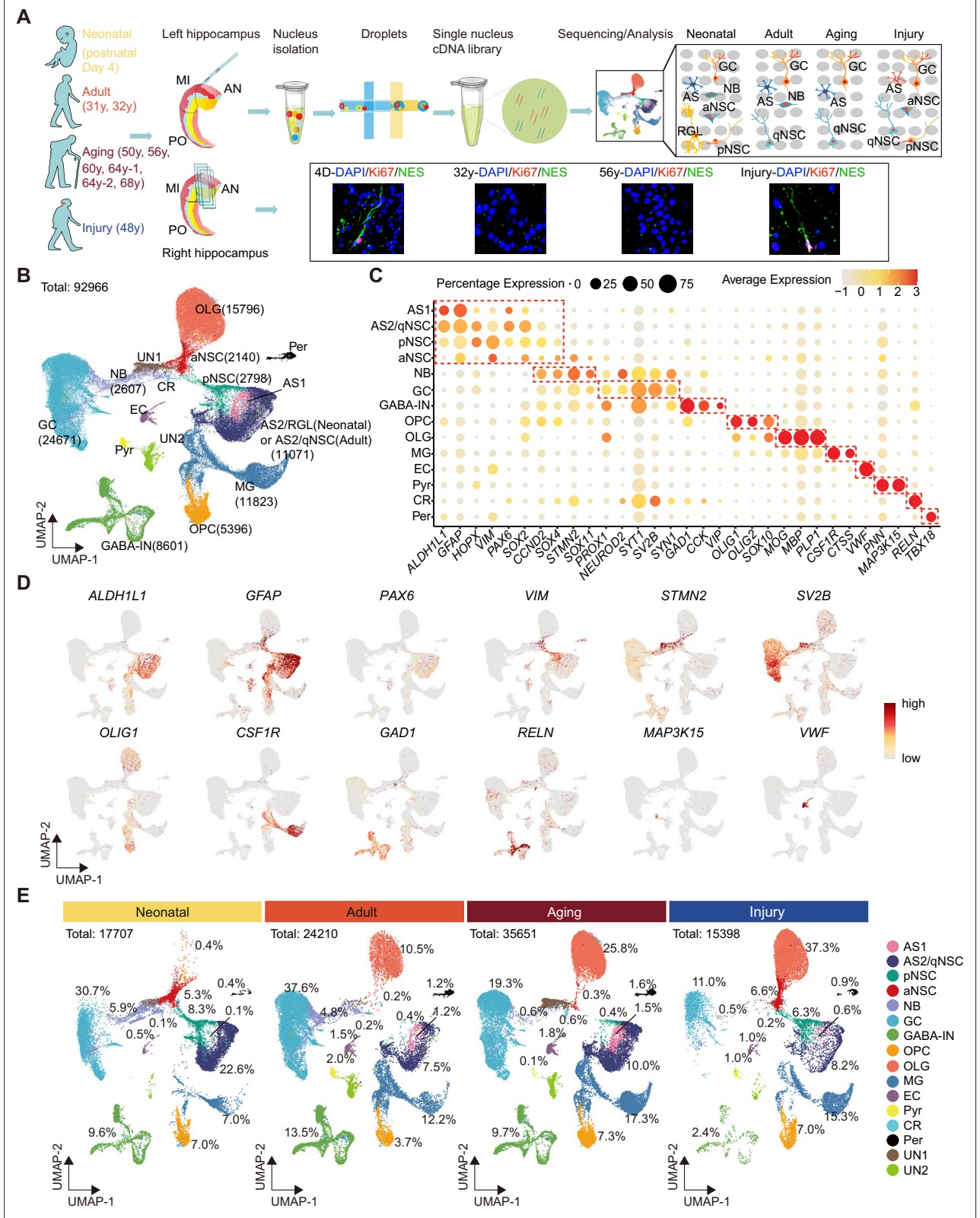

**Figure 1.** Single-nucleus transcriptomic atlas of the human hippocampus across different ages and after stroke injury. (**A**) Summary of the experimental strategy. The pair of hippocampi from post-mortem human donors at different ages were collected. The anterior (AN) and middle (MI) parts containing dentate gyrus were used for single-nucleus RNA-seq (snRNA-seq) and immunostaining. (**B**) 92,966 hippocampal single nuclei were visualized by Uniform Manifold Approximation and Projection (UMAP) plot and categorized into 16 major populations: astrocyte1 (AS1, 1146 nuclei), astrocyte2/quiescent

Figure 1 continued

neural stem cell (AS2/qNSC, 11,071 nuclei), primed NSC (pNSC, 2798 nuclei), active NSC (aNSC, 2140 nuclei), neuroblast (NB, 2607 nuclei), granule cell (GC, 24,671 nuclei), interneuron (IN, 8601 nuclei), pyramidal neuron (PN, 676 nuclei), oligodendrocyte progenitor (OPC, 5396 nuclei), oligodendrocyte (OLG, 15,796 nuclei), microglia (MG, 11,823 nuclei), endothelial cell (EC, 1232 nuclei), pericyte (Per, 981 nuclei), Relin-expressing Cajal-Retzius cell (CR, 218 nuclei), and two unidentified populations (UN1 and UN2, 3810 nuclei). (**C**) Dot plots of representative genes specific for the indicated cell subtypes. The size of each dot represents the cell percentage of this population positive for the marker gene. The scale of the dot color represents the average expression level of the marker gene in this population. (**D**) UMAP feature plots showing expression distribution of cell type-specific genes in all cell populations. Astrocyte (ALDH1L1, GFAP), NSC (PAX6, VIM), neuroblast (STMN2), GC (SV2B), oligodendrocyte progenitor (OLIG1), microglia (CSF1R), interneuron (GAD1, RELN), Relin-expressing Cajal-Retzius cell (RELN), pyramidal neuron (MAP3K15), and endothelial cell (VWF) are shown. Dots, individual cells; gray, no expression; red, relative expression (log-normalized gene expression). (**E**) Quantification of each cell population in the hippocampus at three different age stages and after stroke-induced injury.

The online version of this article includes the following source data and figure supplement(s) for figure 1:

**Source data 1.** Patient information and the expression of findmarker genes used to identify cell populations in Uniform Manifold Approximation and Projection (UMAP).Related to Figure 1.

**Figure supplement 1.** Cell atlas of human hippocampus across different ages and post stoke-induced injury.

stroke-induced injury (48 years of age, n=1) groups (*Figure 1A*, *Figure 1—source data 1*). In total, we sequenced 99,635 single nuclei of which 92,966 nuclei were successfully retained after quality control and filtration. After the removal of cell debris, cell aggregates, and cells with more than 20% of mitochondrial genes transcripts, we analyzed a median of 3001 genes per nucleus. (*Figure 1—figure supplement 1A and B*). To generate an overview of hippocampal cell types, we pooled single cells from all samples and categorized human hippocampal cells based on classical markers and differentially expressed genes (DEGs) into 16 main populations by Uniform Manifold Approximation and Projection (UMAP) (*Figure 1B–D*, *Figure 1—figure supplement 1C–E*). These included astrocyte1 (AS1), astrocytes2/quiescent neural stem cell (AS2/qNSC), pNSC, aNSC, NB, GC, GABA-IN, pyramidal neuron (PN), oligodendrocyte progenitor (OPC), oligodendrocyte (OLG), microglia (MG), endothelial cell (EC), pericyte (Per), Relin-expressing Cajal-Retzius cell (CR), and two unidentified cell types (UN1, UN2). Based on the identified populations, the percentage of each cell population in the hippocampus at three different age stages and after stroke-induced injury was quantified and compared. Although some GCs were lost in the injured hippocampus according to cell percentages, we found that pNSC and aNSC cell numbers decreased markedly with aging but were recovered in the stroke-injured hippocampus (*Figure 1E*). The average number of detected genes in each cell type is similar across different groups, thereby ruling out the possibility that the enrichment of stem cell genes is an artifact of increased global gene expression (*Figure 1—figure supplement 1F*). Overall, cell compositions and proportions varied substantially in neonatal, adult, aging, and injured human hippocampus (*Figure 1E*).

## The heterogeneity and molecular features of human hippocampal NSCs

Since hippocampal neurogenesis is controversial in the adult human brain (*Zhong et al., 2020*; *Cipriani et al., 2018*; *Sorrells et al., 2018*; *Franjic et al., 2022*; *Boldrini et al., 2018*; *Moreno-Jiménez et al., 2019*; *Tobin et al., 2019*; *Terreros-Roncal et al., 2021*) and the dramatic alteration of related cell types at different statuses was observed (*Figure 1E*), we mainly focused on the dissection of NSCs and neurogenic populations. We first performed a cross-species comparison of our human hippocampal neurogenic populations with the published single-cell RNA-seq data from mouse, pig, rhesus macaque, and human hippocampus (*Hochgerner et al., 2018*; *Franjic et al., 2022*). Neurogenic lineage populations across species were distributed at similar coordinates in the UMAP (*Figure 2A*). For example, human AS2/qNSCs and pNSCs aligned more strongly with astrocytes and radial glia-like cells (RGLs) from other species, and expressed classical RGL genes (*Figure 2A*). During embryonic development, pNSCs exhibit greatest similarity to RGLs. However, in the adult stage, pNSCs are in an intermediate state between quiescence and activation. Meanwhile, human aNSCs and NBs clustered together with other species' neural intermediate progenitor cells (nIPCs) and NBs, respectively (*Figure 2A*). In addition, neurogenic lineage markers identified in other species were also highly expressed in corresponding populations in our data (*Figure 2B*).

qNSCs exhibit reversible cell cycle arrest and display a low rate of metabolic activity. However, they still possess a latent capacity to generate neurons and glia when they receive activation signals

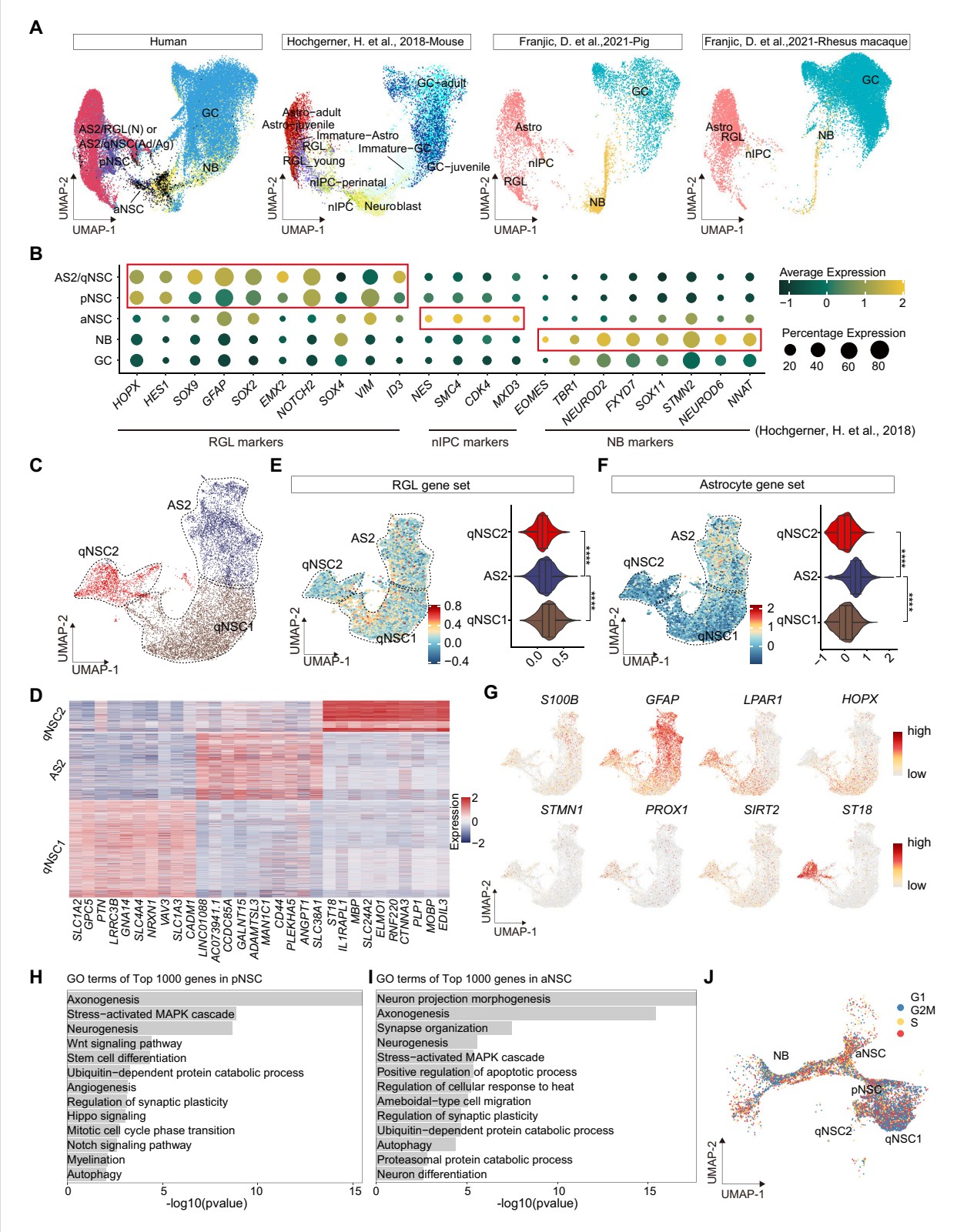

**Figure 2.** Confirmation of neurogenic lineage and dissecting of neural stem cell (NSC) molecular heterogeneity in the postnatal human hippocampus. (**A**) Neurogenic lineage identification was confirmed by cross-species comparison of transcriptomic signatures. Our human data were integrated with published single-nucleus RNA-seq (snRNA-seq) data from mice, pigs, and rhesus macaque by Uniform Manifold Approximation and Projection (UMAP) (***Hochgerner et al., 2018***; ***Franjic et al., 2022***). astrocyte2 (AS2), radial glia-like cell (RGL), neonatal (N), quiescent neural stem cell (qNSC), adult (Ad),

*Figure 2 continued on next page*

*Figure 2 continued*

aging (Ag), primed neural stem cell (pNSC), active neural stem cell (aNSC), neuroblast (NB), granule cell (GC), astrocytes (Astro), neuronal intermediate progenitor cell (nIPC). (**B**) Expressions of previously reported RGL, nIPC, NB, and immature GC markers in the corresponding populations from our human hippocampal single-nucleus RNA-seq (snRNA-seq) data. RGL, radial glia-like cell; nIPC, neural intermediate progenitor cell; NB, neuroblast; and immature GC, immature granule cell. (**C**) The AS2/qNSC population from neonatal sample was subclustered into three clusters, astrocyte2, qNSC1, and qNSC2. (**D**) Heatmap of top 10 genes (p-value <0.05) specific for astrocytes, qNSC1, and qNSC2 after normalization. (**E and F**) Using gene set scores (average, over genes in the set, of Seurat function AddModuleScore) based on previously defined gene sets (*Zamanian et al., 2012*; *Liddelow et al., 2017*; *Clarke et al., 2018*; *Hochgerner et al., 2018*; *Zhong et al., 2020*; *Franjic et al., 2022*) to characterize RGL (**E**) and astrocytes (**F**). Wilcoxon test, the asterisk indicates the p-value < 0.0001. (**G**) UMAP feature plots showing expression distribution of cell type-specific genes. Astrocyte markers (S100B and GFAP), RGL markers (*HOPX* and *LPAR1*), and neuron development markers (*ST18, STMN1, PROX1,* and *SIRT2*) are shown. (**H and I**) Representative gene ontology (GO) terms of the top 1000 genes specifically expressed in pNSCs (**H**) and aNSCs (**I**). (GO:BP, neural development related GO terms, p<0.05). (**J**) Cell cycle phases of qNSC1, qNSC2, pNSC, aNSC, and NB predicted by CellCycleScoring. Each dot represents an individual cell. Steel blue, red, and orange dots represent G1, S, and G2/M phase cells, respectively.

The online version of this article includes the following source data and figure supplement(s) for figure 2:

**Source data 1.** The differential expression genes and related gene ontology (GO) terms of active neural stem cell (aNSC) compared with primed NSC (pNSC).

**Figure supplement 1.** Distinguish quiescent neural stem cells (qNSCs) and astrocytes molecular heterogeneity in the postnatal human hippocampus.

(*Urbán et al., 2019*). They express many genes (i.e. *GFAP, ALDH1L1, VIM*) that are also expressed by astrocytes. Therefore, in our snRNA-seq data, the initial clustering (UMAP) was unable to distinguish qNSCs from astrocytes in the human hippocampus due to their high transcriptional similarity (*Figure 1B*). Previous studies in mice have shown that qNSCs express higher levels of Cd9 and Cd81 than astrocytes (*Llorens-Bobadilla et al., 2015*), and some genes (e.g. Sox9, Hes1, Id4, and Hopx) have been proposed as essential regulators of NSC quiescence (*Zhang et al., 2012*; *Basak et al., 2012*; *Giachino and Taylor, 2014*; *Imayoshi et al., 2010*; *Kawaguchi et al., 2013*; *Zhang et al., 2023*; *Shin et al., 2015*; *Berg et al., 2019*). However, the molecular characteristics of human qNSCs are still not well understood. To investigate the specific features of qNSCs in the human hippocampus, it is crucial to exclude astrocytes from the analysis. To this end, we performed further subclustering of the AS2/qNSC population by using Seurat (FindAllMarker) analysis (*Figure 2C and D*). According to the DEGs and the feature gene expression, three subclusters were identified and annotated as AS2, qNSC1, and qNSC2 (*Figure 2C and D*). Next, we used gene set scores analysis to confirm the properties of AS2, qNSC1, and qNSC2 according to the global gene expression level (*Figure 2E and F*, *Figure 2—figure supplement 1A and B*). Although the RGL gene set hardly distinguishes qNSCs from astrocytes (*Figure 2E*), analysis of astrocyte feature genes (*Zamanian et al., 2012*; *Liddelow et al., 2017*; *Clarke et al., 2018*; *Hochgerner et al., 2018*; *Zhong et al., 2020*; *Franjic et al., 2022*) revealed that the AS2 cluster obtained higher astrocyte score than qNSC1 and qNSC2 (*Figure 2F*). The classical astrocyte markers such as *S100B* and *GFAP* were highly expressed in the AS2 cluster (*Figure 2G*). The qNSC1 cluster was characterized by the preferred expression of quiescence NSC gene *HOPX*. Compared with the qNSC1 cluster, the qNSC2 cluster behaved less quiescent since they highly expressed *LPAR1*, neurogenic genes (e.g. *STMN1, PROX1, SIRT2,* and *ST18*), showing the initial potential of lineage development (*Figure 2D and G*). It is unexpected that we observed high expression of a few OL (oligodendrocyte) genes in cluster qNSC2. However, when we compared the transcriptional similarity of qNSC2 to other populations, we still found a high correlation coefficient between qNSC2 and NSC and astrocyte populations (*Figure 2—figure supplement 1C*). We also observed that the ratio of NSCs in the astroglial lineage clusters remains higher compared to traditional histology studies. However, our data indicate a reduction in qNSCs and an increase in astrocytes during aging and injury, which supports that cell-type identification by using gene set score analysis is effective, although still not optimal. Combined methods to accurately distinguish between qNSCs and astrocytes are required in the future. Compared with astrocyte and qNSCs, pNSCs lowly expressed *ALDH1L1* and *GFAP*, but highly expressed stem cell genes *HOPX, VIM, SOX2, SOX4,* and *CCND2* (*Figure 1C and D*). Consistent with their identities, gene ontology (GO) terms of the top 1000 genes in pNSCs included stem cell differentiation, Wnt signaling, neurogenesis, Notch signaling, and hippo signaling, indicating that they maintain critical properties of RGLs (*Figure 2H*). Different from pNSCs, the identified aNSCs highly expressed stem cell and proliferation markers, such as *SOX2, SOX4, SOX11,* and *CCND2* (*Figure 1C and D*) and were enriched for GO terms associated with the

onset of neuronal fate, such as neuron differentiation, neuron projection morphogenesis, axonogenesis, and synapse organization (*Figure 2I*). When we compare the DEGs between pNSC and aNSC (*Figure 2—figure supplement 1D*, *Figure 2—source data 1*), we also found that pNSC is more associated with the Wnt signaling pathway, axonogenesis, and Hippo signaling, while aNSC is more associated with G2/M transition of mitotic cell cycle, neuron projection development, axon development, and dendritic spine organization (*Figure 2—figure supplement 1E*, *Figure 2—source data 1*). Thus, the pNSCs referred to in this study represent an intermediate state between quiescence and activation. Different from NSCs, NBs highly expressed *CCND2*, *SOX4*, *STMN2*, *SOX11*, *PROX1*, and *NEUROD2*, and started to express several GC markers, such as *SYT1* and *SV2B* (*Figure 1C and D*). As expected, qNSC1 and qNSC2 were mainly in the non-cycling G0/G1 phase whereas aNSCs were mainly in the S/G2/M phase of active mitosis, confirming their quiescent and active cell states, respectively (*Figure 2J*).

Taken together, our findings demonstrate the molecular features of various types of human hippocampal NSCs and their progeny, including qNSCs, pNSCs (RGLs), aNSCs, and NBs, highlighting the heterogeneity of these cell populations and their unique cell cycle properties.

## Novel markers distinguishing various types of NSCs and NBs in the human hippocampus

The lack of validated cell type-specific markers constrains efforts to identify NSCs and their progeny in the human hippocampus. Since single-cell hierarchical Poisson factorization (scHPF) (*Levitin et al., 2019*) could sort out specific genes and Seurat analysis (FindAllMarkers) is suitable for searching highly expressed genes, we used the two methods together to narrow the scope of candidate genes, allowing us to identify specific genes that can distinguish qNSCs, pNSC (RGL), aNSC, and NB cells from other non-neurogenic cells in the human hippocampus. The combined results from scHPF and FindAllMarkers data showed that *LRRC3B*, *RHOJ*, *SLC4A4*, *GLI3* were specifically expressed in qNSC1 and qNSC2, *CHI3L1*, and *EGFR* could be regarded as pNSC marker genes, and *NRGN*, *NRN1*, and *HECW1* as NB marker genes at the transcriptional level (*Figure 3A–C*, *Figure 3—source data 1*). Feature plots revealed that *EGFR* was specific for pNSCs, while *CHI3L1* was also expressed by astrocytes. *NRGN* and *NRN1* but not *HECW1* were specific for NBs (*Figure 3C*). Notably, several genes enriched in NBs, such as *HECW1*, *STMN2*, *NSG2*, *SNAP25*, and *BASP1* (*Figure 3C*), were also widely distributed in GABA-INs that were validated by high expression of known GABA-IN marker genes, such as *DLX1*, *GABRG3*, *CCK*, *SLC6A11*, *SLC6A1*, *GAD1*, *GAD2*, *CNR1*, *GRM1*, *RELN*, and *VIP* (*Figure 3C and D*). When we compared the GC lineage and the interneuron population at the whole transcriptome level between our dataset and published mouse (*Hochgerner et al., 2018*), macaque and human (*Franjic et al., 2022*) transcriptome datasets, we found high transcriptomic congruence across different datasets (*Figure 3—figure supplement 1A*). Specifically, our identified human GABA-INs very highly resembled the well-annotated interneurons in different species (similarity scores >0.95) (*Figure 3—figure supplement 1A*). Based on the validated cell-type annotation, we plotted expression of the NB/im-GC highly expressed genes reported by the other studies (*Zhou et al., 2022*; *Wang et al., 2022*) in our identified GC lineage and interneuron population (*Figure 3D*). Indeed, both the previously reported genes that are regarded as the markers of NB/im-GCs, such as *DCX*, *PROX1*, and *CALB2* (*Zhou et al., 2022*; *Wang et al., 2022*), and here identified genes (*HECW1*, *STMN2*, *NSG2*, *SNAP25*, and *BASP1*) were also enriched in neonatal GABA-INs (*Figure 3C and D*). Consistently, these genes were also prominently expressed in the adult interneurons (*Figure 3—figure supplement 1B*). To confirm the protein expression of DCX in interneurons, we conducted co-immunostainings of DCX and a typical interneuron marker (SST). Our results demonstrate that SST-positive interneurons are indeed capable of being stained by the traditional NB marker DCX in primates (*Figure 3—figure supplement 2A—C*). These results suggest that identification of newborn neurons using NB/im-GC genes requires the exclusion of the interneuron contamination, as reported by a recent study (*Franjic et al., 2022*).

To further identify NB/im-GC-specific genes absent in interneurons, we mapped the NB/im-GC genes identified by scHPF (top 100) and FindAllMarkers (p-adjust <0.01) onto the interneuron population (*Figure 3E*). We selected genes with low or absent expression in the interneuron population (around the coordinate origin) as NB/im-GC-specific genes by filtering out genes with high and wide distributions in the interneuron population (*Figure 3E*). We identified several representative NB/

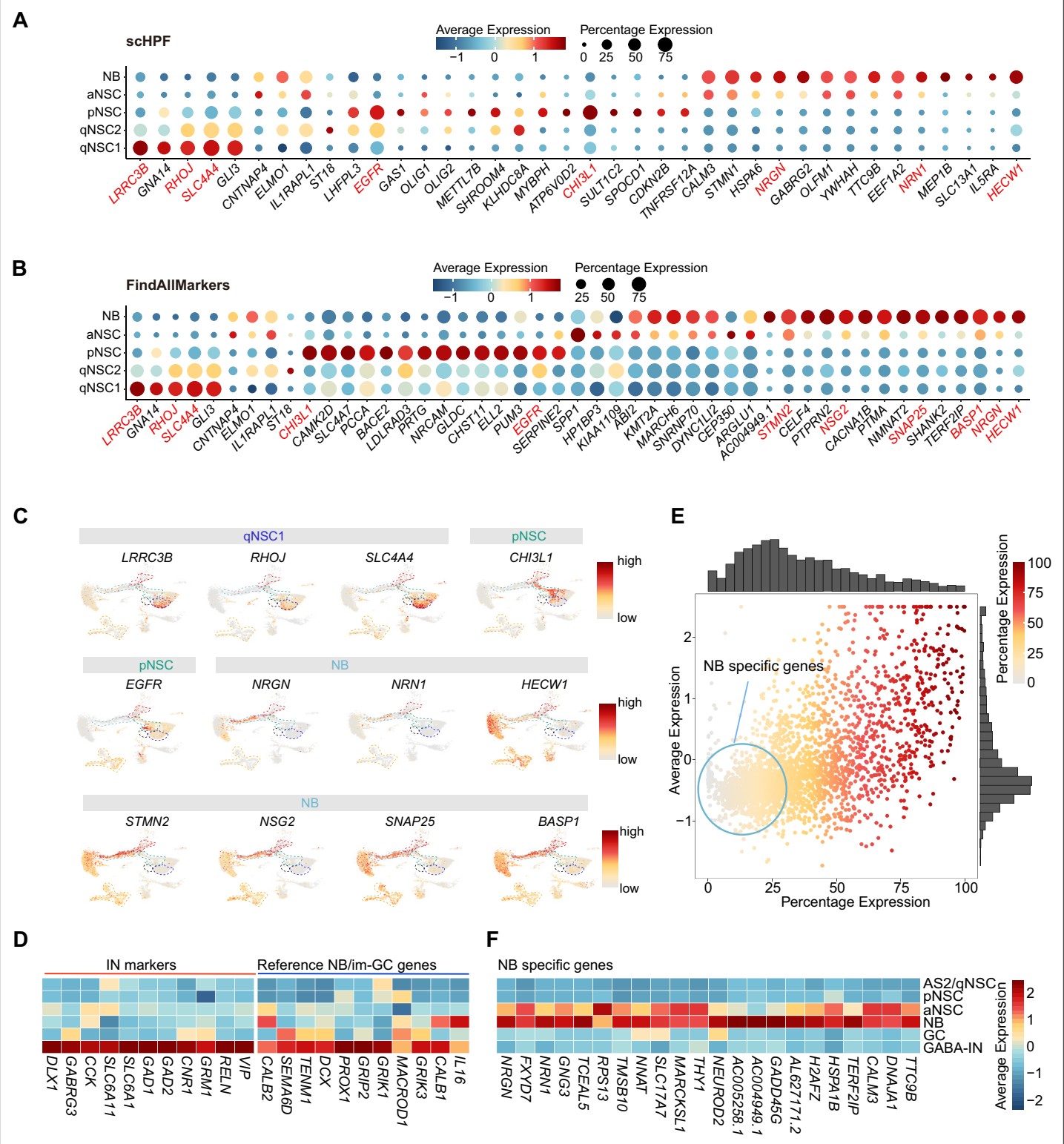

**Figure 3.** Discovery of novel markers distinguishing various types of neural stem cells (NSCs) and neuroblasts (NBs) in the human hippocampus. (**A and B**) Representative top genes specific for qNSC1, qNSC2, pNSC, aNSC, and NB in the neonatal neurogenic lineage identified by single-cell hierarchical Poisson factorization (scHPF) (**A**) and FindAllMarkers function of Seurat (**B**). (**C**) Uniform Manifold Approximation and Projection (UMAP) visualization of several cell type-specific genes of the qNSCs, pNSC, and NB predicted by scHPF and FindAllMarkers. (**D**) Heatmap showing that neuroblast/immature GC highly expressed genes that are previously reported by other literature were widely expressed in human hippocampal interneurons. (**E**) Scatter plot showing that several NB genes predicted by scHPF and findmarker from our single-nucleus RNA-seq (snRNA-seq) data were also widely expressed in

*Figure 3 continued on next page*

*Figure 3 continued*

human hippocampal interneurons. The genes without/with low expression in the interneurons were selected as NB-specific markers (red circle scope). (**F**) NB-specific genes selected from our snRNA-seq data were not or very lowly expressed in astrocytes2 (AS2)/qNSCs, pNSC, GC, and interneurons.

The online version of this article includes the following source data and figure supplement(s) for figure 3:

**Source data 1.** Potential marker genes identified by FindAllMarker and single-cell hierarchical Poisson factorization (scHPF).

**Figure supplement 1.** Reported neuroblast genes were widely distributed in the adult human interneurons.

**Figure supplement 2.** The neuroblast marker DCX was expressed in interneurons (SST+) in the hippocampus of 3-month-old macaques.

im-GC-specific genes, such as *CALM3, TTC9B, NRGN, FXYD7, NRN1, GNG3, TCEAL5, TMSB10,* and *NEUROD2* (*Figure 3F*) and confirmed their specificity in adult and aging samples (*Figure 3—figure supplement 1C*).

Overall, our results revealed that most NB/im-GC genes are prominently expressed in human hippocampal interneurons, hence, our newly identified NB marker genes could be used to identify newborn neurons in adult or aging human hippocampus.

## The developmental trajectory and molecular cascades of NSCs in neonatal human hippocampus

Based on studies in mice, RGLs acquire quiescence gradually throughout postnatal development and adulthood, and share molecular markers with astrocytes (*Berg et al., 2019*; *Alvarez-Buylla and Garcia-Verdugo, 2002*; *Hochgerner et al., 2018*; *Ponti et al., 2013*; *Seri et al., 2001*; *Steiner et al., 2004*; *Garcia et al., 2004*). The situation of RGLs in human hippocampus is still unclear. We used RNA velocity to investigate the developmental potentials of NSCs in the neonatal human hippocampus (*Figure 4A*). We observed that pNSCs give two developmental directions: one is entering quiescence or generating AS2, and the other is generating aNSCs. Based on the GO term analysis of the DEGs comparing qNSC1/2 with pNSCs, it appears that pNSCs are more active than qNSCs (*Figure 4B*). Since qNSCs originate from RGLs (*Figure 4A*) but exit out of the cell cycle and development, the pNSC (RGL) population was set as the root of the developmental trajectory to recapitulate the continuum of the neurogenesis process (*Figure 4C*). According to the developmental trajectory, pNSCs were followed by aNSCs and NBs, and gave rise to two types of neurons (N1 and N2), indicative of ongoing neurogenesis (*Figure 4C* and *Figure 4—figure supplement 1A and B*). The N1 and N2 populations had distinct gene expression profiles, which indicates they are subtypes of GCs (*Figure 4—figure supplement 1C*). The N1 specifically express *NCKAP5, SGCZ, DCC, FAM19A2,* whereas the N2 specifically express *FLRT2, RIMS2, NKAIN2,* and *XKR4* (*Figure 4—figure supplement 1D*).

Next, we traced the dynamics of pNSC, aNSC, NB, and GC marker genes along with the developmental trajectory (*Figure 4D*). We found that *HOPX, SOX2,* and *VIM* expression was preferentially maintained in pNSCs and aNSCs, but decreased upon differentiation. Finally, expressions of NB genes *DCX* and *STMN2* increased along the trajectory and the GC gene *SYT1* reached maximum expression at the end of the trajectory (*Figure 4D*). To validate the RNA-seq results, we performed immunostaining in the DG of D4 neonatal hippocampus. In the granule cell layer (GCL) and hilus, we detected HOPX+NES+ RGLs and Ki67+NES+ proliferating NSCs (*Figure 4E*), consistent with previously reported NSC immunostaining in human hippocampus (*Sorrells et al., 2018*). We also found that PSA-NCAM+ NBs located in the GCL as clusters (*Figure 4E*). These results confirm that both pNSCs (RGLs) and aNSCs maintain their proliferative status and can generate new GCs in the neonatal human hippocampus.

To understand how gene expression profiles in different cell populations change over the developmental trajectory, we constructed gene expression cascades of neurogenesis-related cell populations (including pNSC, aNSC, NB, and N1/N2) and annotated the DEGs into four clusters (*Figure 4F*). Cluster 1 population, located at the trajectory start, consists of pNSCs and aNSCs with high expression of *VIM, GFAP, SOX6, GPC6, CD44, CHI3L1, TNC, EGFR,* and *HOPX*. These genes are mainly related to cell proliferation, regeneration, angiogenesis, and the canonical Wnt signaling pathway (*Figure 4F*, *Figure 4—source data 1*, *Figure 4—figure supplement 2*). As expected, *HOPX, VIM, CHI3L1,* and *TNC* were down-regulated along the presudotime (*Figure 4G*). Cluster 2 population highly expressed neurogenesis and neuronal differentiation genes, including *NRGN, STMN2,* and

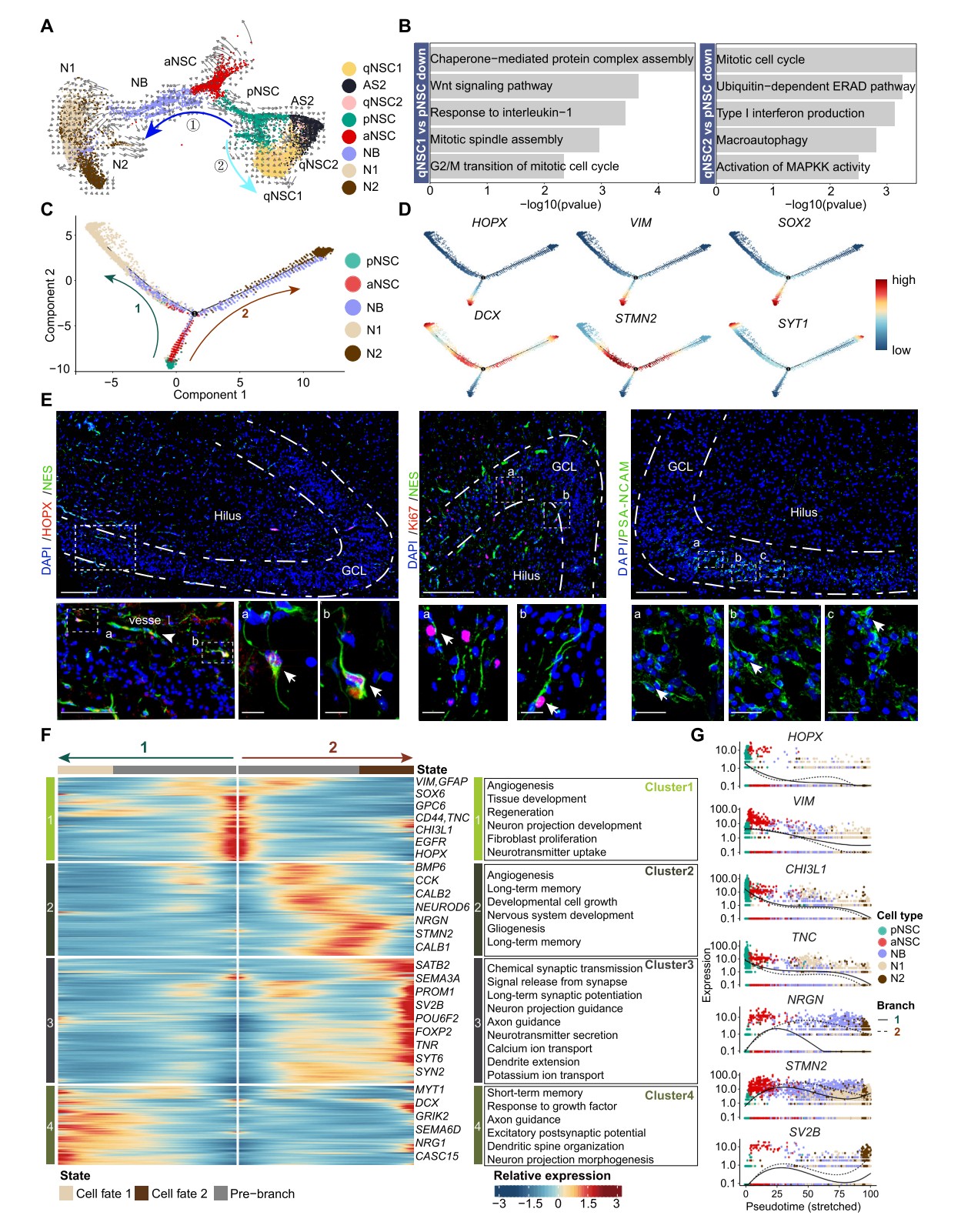

**Figure 4.** The transcriptional dynamics predicated by RNA velocity and pseudotime reconstruction revealed developmental potentials of neural stem cell (NSC) in the neonatal human hippocampus. (**A**) RNA velocity analysis indicating the developmental trajectory of hippocampal neurogenic lineage at postnatal day 4. Cell types are labeled. (**B**) Representative gene ontology (GO) terms of the differentially expressed genes compare qNSC1, qNSC2 with primed NSC (pNSC). (**C**) Pseudotime reconstruction of the neurogenic lineages in the neonatal human hippocampus. Dots showing individual cells.

*Figure 4 continued on next page*

*Figure 4 continued*

Different color represents different cell types. The arrows indicate the directions of differentiation trajectories. pNSCs as the development root was successively followed by active NSCs (aNSCs) and neuroblasts, and then separated into two branches (1 and 2), generating two types of neuronal cells N1 and N2, respectively. (**D**) Expression dynamics of cell type-specific genes along with the pseudotime. Each dot represents an individual cell. NSC genes (*HOPX, VIM,* and *SOX2*), granule neuroblast genes (*DCX* and *STMN2*), and mature granule cell gene (*SYT1*) are shown. (**E**) Immunostainings of radial glia (NSC) markers (HOPX and NES), aNSC markers (NES and Ki67), and neuroblast marker (PSA-NCAM). The HOPX+NES+ RGLs and NES+Ki67+ active NSCs with long apical processes were detected in postnatal day 4 hippocampal dentate gyrus (arrows). The PSA-NCAM+ neuroblasts (green) were located across the granule cell layer (GCL). Scale bars of HOPX/NES immunostaining are 200 μm; the magnified and further magnified cell images are 100 μm and 10 μm, respectively; the arrowhead indicates the vessel. Scale bars of KI67/NES immunostaining are 100 μm and 10 μm, respectively. Scale bars of PSA-NCAM immunostaining are 100 μm and 10 μm, respectively; arrows indicate the neuroblasts. (**F**) Heatmap showing that differentially expressed genes along the pseudotime were divided into four clusters. Representative genes and enriched gene ontology (GO) terms of each cluster are shown (GO:BP, neural development related GO terms, p<0.05). (**G**) Representative NSC genes (*HOPX, VIM, CHI3L1,* and *TNC*) and neuronal genes (*NRGN, STMN2,* and *SV2B*) were ordered by Monocle analysis along with the pseudotime. Cell types along with the developmental trajectory were labeled by different colors.

The online version of this article includes the following source data and figure supplement(s) for figure 4:

**Source data 1.** Genes and enriched gene ontology (GO) terms of *Figure 4B and F*.

**Figure supplement 1.** Pseudotime reconstruction of the neurogenic lineage development in the neonatal day 4 human hippocampus.

**Figure supplement 2.** Differentially expressed genes along the pseudotime of neurogenic lineage in the neonatal human hippocampus.

*NEUROD6*, which reached their expression peak at the middle of the trajectory and represented neuron development (*Figure 4F and G*, *Figure 4—source data 1*). Cluster 3 and 4 populations, respectively located at the end of two branches, contained neurons that became mature and functional. The genes for the branch 2 (cluster 3) were associated with axon guidance, neurotransmitter secretion, long-term synaptic potentiation, and ion transport, such as *SV2B, SYT6,* and *SYN2*; similarly, the branch 1 (cluster 4) genes were associated with neuron projection guidance, dendritic spine organization, and excitatory postsynaptic potential, such as *MYT1* and *GIRK2* (*Figure 4F*). In addition, we also identified transcription factors (TFs) that are differentially expressed from pNSC to neurons along with the neurogenesis trajectory in all four clusters (*Figure 4—figure supplement 1E*). For example, progenitor cell regulation TFs *PBX3, PROX1, GLIS3, RFX4,* and *TEAD1* were dominantly enriched in the origin of the trajectory. Conversely, differentiation-related TFs *POU6F2, FOXP2, THRB, ETV1, NR4A3, BCL11A, NCALD, LUZP2,* and *RARB* were prominently expressed in the middle and the end of the trajectory.

Our findings collectively imply that specific types of human hippocampal NSCs remain in a quiescent state postnatally, serving as a reservoir for potential neurogenesis. Meanwhile, a considerable proportion of NSCs retain their capacity for proliferation and can produce fresh GCs in the neonatal hippocampus of humans.

## Most NSCs become deep quiescence during aging

When we quantified the cell numbers of different types of NSCs and their progeny across neonatal (postnatal day 4), adult (the mean of 31y, 32y), and aging (the mean of 50y, 56y, 60y, 64y-1, 64y-2, 68y) groups, we noted that the ratios of qNSC1 and qNSC2 increased significantly with age, particularly the deep quiescent stem cell qNSC1. Conversely, pNSC and aNSC populations sharply declined from neonatal to adult and aging stages. Meanwhile, the numbers of NBs were comparable in the neonatal and adult brain, but they were markedly reduced in the aging hippocampus (*Figure 5A and B*). The abundance of NBs in the adult brain suggest that compared to rodents, immature neurons in primates are indeed retained for a longer period and possess the potential to further develop into mature neurons (*Kohler et al., 2011*). Although the number of these neurogenic cells (pNSCs, aNSCs, NBs) in the aged hippocampus is quite low, they still expressed NSC and NB marker genes, including *VIM, PAX6, SOX2, PROX1, NRGN, INPP5F,* and *TERF2IP* (*Figure 5—figure supplement 1A*), ruling out that these are contaminated by other neurogenic cell types. These results together showed that pNSCs and aNSCs are present in the neonatal hippocampus, but their numbers significantly decrease with age. This suggests that human neurogenesis experiences a rapid decline after birth. In contrast, NBs have a longer maturation period until adulthood, which is consistent with previous studies (*Ayhan et al., 2021*; *Franjic et al., 2022*; *Ngwenya et al., 2015*; *Seki, 2020*).

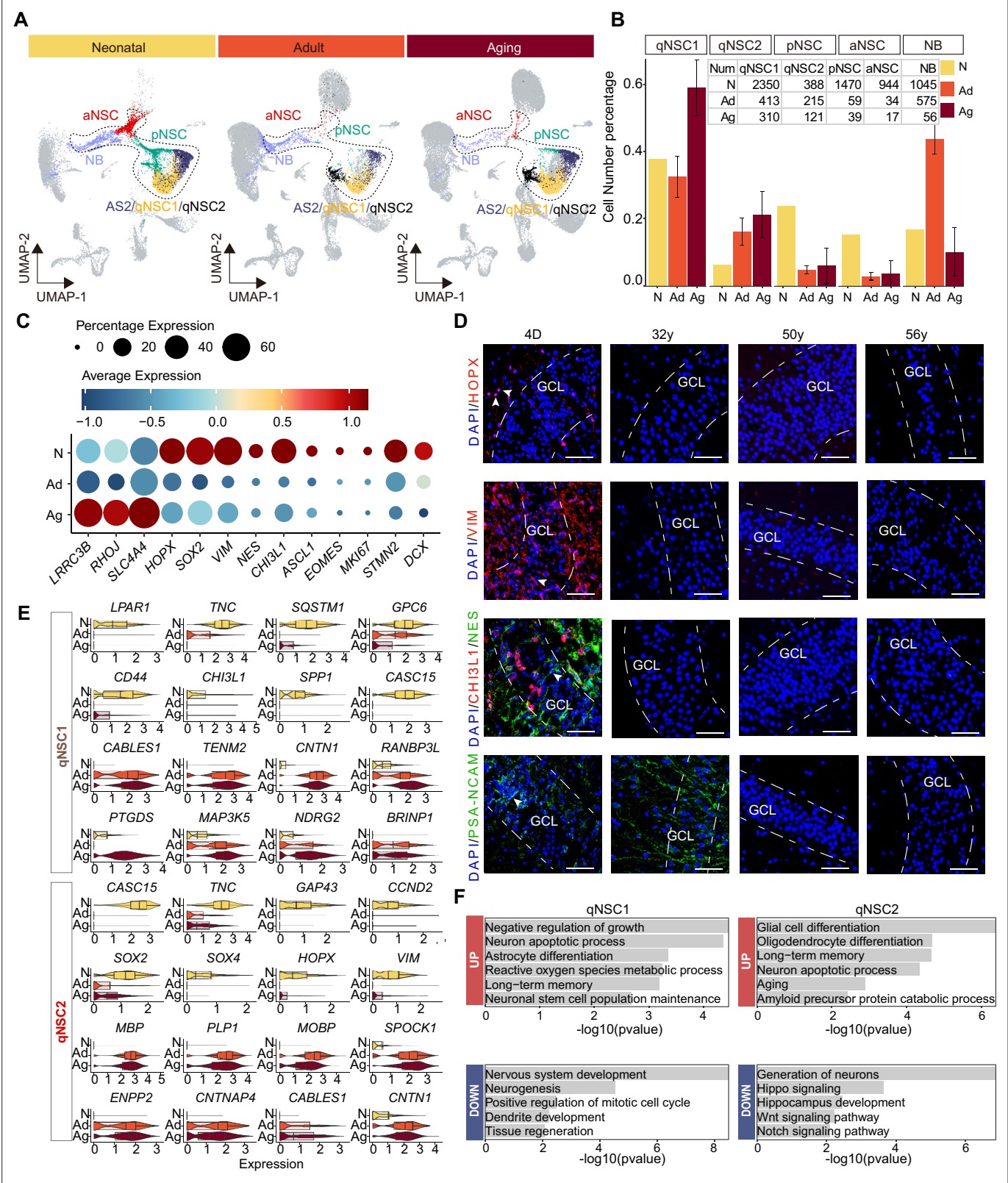

**Figure 5.** Age-dependent molecular alterations of the hippocampal neural stem cells (NSCs) and neuroblasts (NBs). (**A and B**) Feature plots (**A**) and quantification (**B**) of the neurogenic populations during aging. Neonatal (abbreviated as N), adult ( abbreviated as Ad), aging ( abbreviated as Ag). The neurogenic populations include qNSC1, qNSC2, pNSC, aNSC, and neuroblast. (**C**) The dynamic expression of some representative genes, including newly identified qNSCs genes (*LRRC3B, RHOJ,* and *SLC4A4*), NSC genes (*HOPX, SOX2, VIM, NES,* and *CHI3L1*), neural progenitor or proliferation genes

*Figure 5 continued on next page*

*Figure 5 continued*

(*ASCL1, EOMES,* and *MKI67*), and immature granule cell genes (*STMN2* and *DCX*), in human hippocampus across neonatal (postnatal day4), adult (31y, 32y), and aging (50y, 56y, 60y, 64y-1, 64y-2, 68y). (**D**) Immunostaining of classical NSC markers (HOPX, VIM, and NES) in human hippocampal dentate gyrus across different ages (postnatal day 4, 32y, 50y, 56y). Scale bars, 60 μm. The arrowheads indicate positive cells with typical morphology. (**E**) Violin plot showing differentially expressed genes of qNSC1 and qNSC2 in the aging group compared to the neonatal group. (**F**) Representative gene ontology (GO) terms of significantly (p-value <0.05) up- and down-regulated genes in qNSC1 and qNSC2 during aging.

The online version of this article includes the following source data and figure supplement(s) for figure 5:

**Source data 1.** Genes and enriched gene ontology (GO) terms of qNSC1, qNSC2, primed neural stem cell (pNSC), active NSC (aNSC), and neuroblast (NB) populations during aging.

**Figure supplement 1.** Alterations of the neurogenic lineage in human hippocampus during aging.

**Figure supplement 2.** Differentially expressed genes (DEGs) and enrichment functions in primed neural stem cell (pNSC), active NSC (aNSC), and neuroblast (NB) along aging, respectively.

Given the continuity of NSC development and the rarity of NSCs in the adult or aged hippocampus, we merged the five cell types qNSC1s, qNSC2s, pNSCs, aNSCs, and NBs together as the neurogenic lineage to analyze the transcriptomic alterations during aging. We have observed a significant up-regulation of astrocyte and quiescence genes (*LRRC3B, RHOJ, SLC4A4*) with increasing age, as well as a marked down-regulation of pNSC genes (*HOPX, SOX2, NES, VIM,* and *CHI3L1*), aNSC genes (*ASCL1, EOMES,* and *MKI67*), and NB genes (*STMN2, DCX*) upon aging (*Figure 5C*). When we stained hippocampal tissue sections from neonatal D4, 32y, and 56y donors (*Figure 5D* and *Figure 5—figure supplement 1B*), we observed that NSC markers HOPX, VIM, NES, and CHI3L1, which were widely expressed in the neonatal D4 DG, were almost lost in 32y, 50y, and 56y DG. VIM⁺ and NES⁺ RGLs were only present around the GCL or in the hilus of the neonatal DG (D4); whereas the NB marker PSA-NCAM was expressed in the D4 and 32y DG, but not in 50y and 56y hippocampus (*Figure 5D*). In agreement with previous staining in adult human brain samples (*Sorrells et al., 2018*), PSA-NCAM⁺ cells in the GCL of the neonatal and adult DG had neuronal morphologies (*Figure 5D*). Together, our immunostaining analysis is consistent with our snRNA-seq data, confirming that pNSCs and aNSCs experience a significant loss with aging, while NBs are sustained until adulthood in humans.

To explore whether the human hippocampal NSCs are getting more and more quiescent during postnatal development and aging, we compared qNSCs from the neonatal sample with those from aged samples (*Figure 5—source data 1*). We observed that cell proliferation and growth inhibition genes (*BRINP1, CABLES1, TENM2, CNTN1*), and stem cell differentiation genes (*RANBP3L, NDRG2*) were up-regulated significantly in qNSC1 during aging. Besides *CABLES1* and *CNTN1*, the oligodendrocyte genes (*MBP, PLP1, MOBP*) were also highly expressed in aging qNSC2 (*Figure 5E*). In contrast, stem cell and regeneration genes (*LPAR1, TNC, CASC15, SOX2, SOX4, HOPX, VIM*) were down-regulated in qNSC1 and qNSC2 (*Figure 5E*). The enriched GOs of significantly up-regulated genes in aging qNSC1 and qNSC2 included negative regulation of growth, neuronal stem cell population maintenance, astrocyte differentiation, oligodendrocyte differentiation, aging, and amyloid precursor protein catabolic process (*Figure 5F* and *Figure 5—source data 1*). Instead, the enriched GOs of significantly down-regulated genes in qNSC1 and qNSC2 were related to nervous system development, neurogenesis, positive regulation of mitotic cell cycle, tissue regeneration, autophagy, generation of neurons, hippo signaling, Wnt signaling pathway, and Notch signaling pathway (*Figure 5F* and *Figure 5—source data 1*). All these differences between neonatal and aged qNSCs suggest that hippocampal NSCs undergo a transition into a state of deep quiescence and acquire glial properties during aging. In addition, we also compared gene expression of the remaining pNSCs, aNSCs, and NBs across neonatal, adult, and aged groups, respectively (*Figure 5—figure supplement 2A—I*). The DEGs and enriched GOs of each cell type also strongly revealed that neurogenesis decline with aging is mainly due to repression of NSC proliferation, deficient autophagy and proteasomal protein catabolic process and increased glial cell differentiation. Overall, the results obtained from both the comparison of the entire neurogenic lineage and the comparison of individual cell types suggest that most NSCs lose their neurogenic potential as a result of entering a state of deep quiescence during aging.

## Injury-induced activation of qNSCs in the adult hippocampus

The homeostasis of NSCs impacts the dynamics of neurogenesis in response to environmental signals (*Chaker et al., 2015*; *Daynac et al., 2014*; *Enwere et al., 2004*; *Katsimpardi et al., 2014*) and injury conditions in mice can even reactivate qNSCs into a proliferative state that gives rise to new neurons (*Llorens-Bobadilla et al., 2015*; *Buffo et al., 2008*). In the stroke-afflicted donor (48y), we noted a significant loss of hippocampal granule neurons and interneurons (*Figure 1E*). Compared to adult donors, genes associated with apoptosis, DNA damage, and autophagy were significantly up-regulated in the GCs and GABA-INs of the stroke-injured hippocampus (*Figure 6—figure supplement 1A and B*). Consistently, we detected evident cell apoptosis by terminal deoxynucleotidyl transferase dUTP nick end labeling (TUNEL) assay in the stroke-injured DG, but not in other adult or aging samples (*Figure 6—figure supplement 1C*). These data validated that injury had occurred in the hippocampus of donors that had suffered from a stroke. Interestingly, we observed that pNSCs and aNSCs are predominantly present in the neonatal and stroke-injured samples, with minimal presence in other groups. Meanwhile, qNSCs increased with aging and reduced upon injury (*Figure 6A*). These results indicated that qNSCs may be reactivated upon injury and give rise to pNSC and aNSC populations. However, we only observed very few cells in the NB population that highly expressed NB marker genes *PROX1, SEMA3C, TACC2, INPP5F,* and *TERF2IP* (*Figure 5—figure supplement 1A*). We speculate that because the patient died within 2 days after the stroke, there was little time for the activated NSCs to generate more NBs.

Previous studies in mice reported that NSCs and astrocytes become activated after stroke around the injured area. Such activated NSCs which could generate newborn neurons together with reactive astrocyte-formed glial scarring may contribute to brain repair (*Benner et al., 2013*; *Faiz et al., 2015*; *Li et al., 2010*). Since activated NSCs and reactive astrocytes share similar transcriptional properties but have distinct morphology, we performed immunostaining of NES/KI67, NES/VIM, and NES/CHI3L1 in the stroke-injured DG. We detected a few NES⁺KI67⁺, NES⁺VIM⁺, and NES⁺CHI3L1⁺ aNSCs that had radial glia morphology with apical processes (*Figure 6B*), appearing similar to D4 NES⁺ RGLs (*Figure 4E*). However, we could not detect these aNSCs in any other adult sample (32y, 50y, 56y) (*Figure 5D*). Since VIM⁺CHI3L1⁺ reactive astrocytes with an irregular contour or star-shape morphology were widely observed in the injured hippocampus (*Figure 6—figure supplement 1E*), the pNSC and aNSC populations identified through initial UMAP clustering may include reactive astrocytes. To distinguish activated NSCs (pNSCs and aNSCs) from reactive astrocytes, we integrated neonatal pNSCs and aNSCs with injury samples, and then applied neonatal pNSC and aNSC as cell prototypes to identify pNSCs and aNSCs in the injury sample. We increased cluster resolution and obtained eight subclusters with distinct gene expression profiles (*Figure 6—figure supplement 2A and B*). When we compared the fraction of each subcluster in neonatal and injury samples, we found subclusters 0, 1, and 3 were predominant in the neonatal sample, and subclusters 2 and 4 were predominant in injury sample (*Figure 6—figure supplement 2C*). The results of gene set score analysis also showed that subclusters 0, 1, and 3 maintained higher RGL potential than subcluster 4, and subcluster 2 had more evident reactive astrocyte properties than subclusters 0, 1, and 3 (*Figure 6C*). Consistently, RGL-specific genes (*VIM, HOPX, LPAR1,* and *SOX2*) were significantly expressed in subcluster 0, 1, and 3. The neurogenic genes (*STMN1, DCX,* and *SIRT2*) were mainly expressed in subcluster 0. The reactive astrocyte marker (*OSMR, TIMP1,* and *LGALS3*) were mainly expressed in subcluster 2 (*Figure 6—figure supplement 2D*). Therefore, cells in subcluster 0 were speculated as pure aNSCs, subclusters 1 and 3 were pNSC in the stroke-injured hippocampus, and cells in subcluster 2 were reactive astrocytes. Since the features of other small subclusters were not clear, they were excluded from the subsequent developmental trajectory analysis. When we quantified qNSC1, qNSC2, pNSCs, aNSCs and reactive astrocytes in neonatal (N), adult (Ad), aging (Ag), and stroke-injured (I) hippocampus, we still found that the ratios of the pNSC and aNSC populations in the neurogenic lineage reached up to 17.3% (322/1861) and 11.7% (218/1861), comparable to ratios in the neonatal group and significantly higher than the adult and aging groups. Correspondingly, the ratios of qNSC populations qNSC1 (23.4%, 413/1258) and qNSC2 (11.1%, 214/1258) in the neurogenic lineage evidently decreased in the injury group compared with adult (qNSC1=32.8%, 413/1258; qNSC2=17.0%, 214/1258) and aged group (qNSC1=57.8%, 310/536; qNSC2=22.6%, 121/536) (*Figure 6D*). These results together with the decline of neurogenesis in the aging group suggest that some qNSCs in the adult and aging human hippocampus can be reactivated and give rise to aNSCs upon stroke-induced injury.

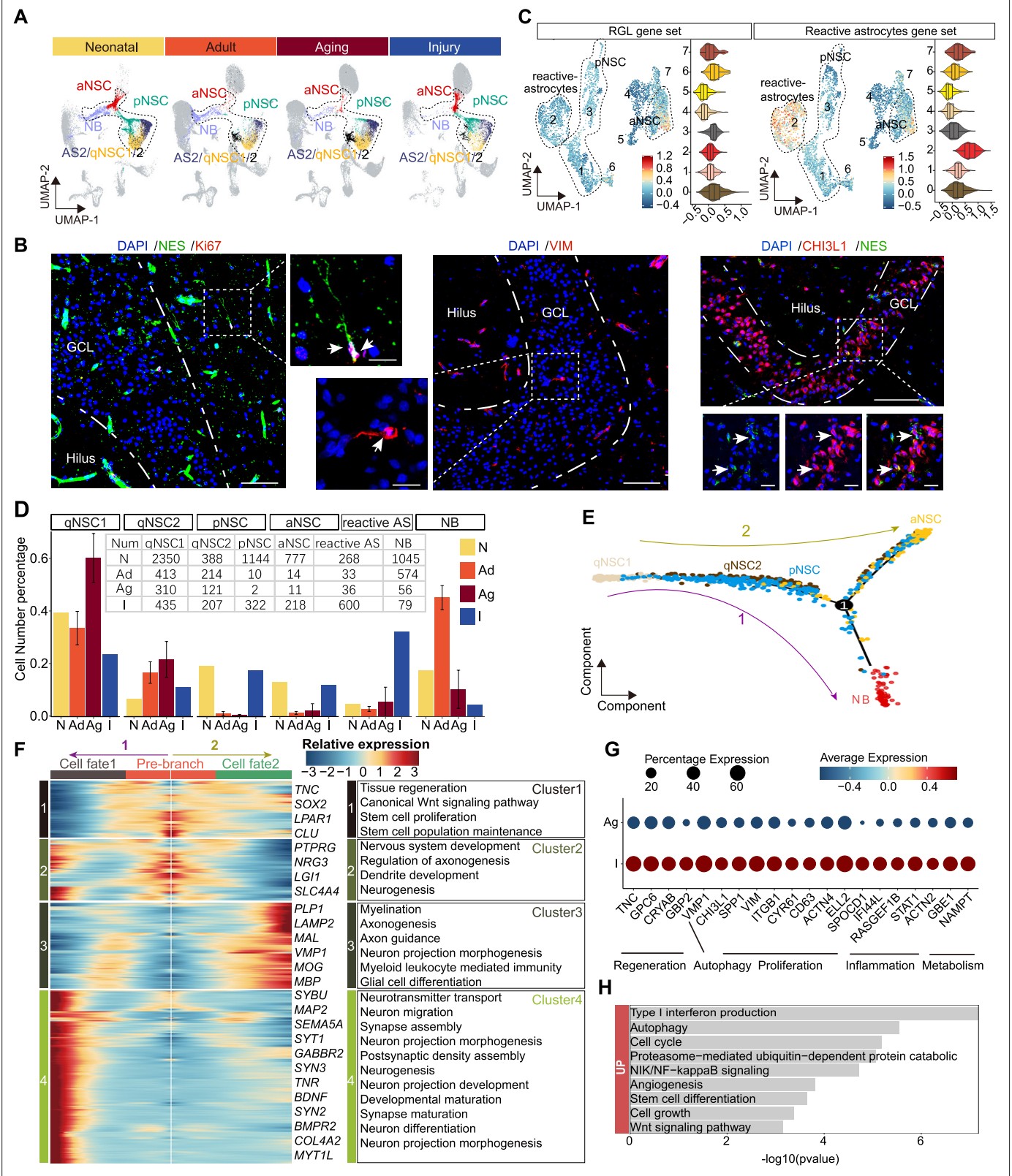

**Figure 6.** The transcriptomic signatures of the activated neurogenic lineage in the adult human injured hippocampus induced by stroke. (**A**) The neurogenic lineage included qNSC1, qNSC2, reactivated primed/active neural stem cell (pNSC/aNSC) and neuroblast (NB). Cell distribution showing by feature plots. (**B**) Immunofluorescence images of NES (green)/Ki67 (red), VIM (red), and CHI3L1 (red)/NES (green) showing a few active NSCs in the 48-year-old injured hippocampal dentate gyrus. The arrows indicate radial morphology NES+/KI67+, VIM+, or CHI3L1+/NES+ active NSCs,

*Figure 6 continued on next page*

*Figure 6 continued*

respectively. Scale bars, 100 µm; the magnification, 20 µm. (**C**) Annotated into pNSC, aNSC, and reactive astrocytes according to gene set scores (average, over genes in the set, of Seurat function AddModuleScore). (**D**) Quantification of qNSC1, qNSC2, pNSCs, aNSCs, reactive astrocytes, and NB in neonatal (abbreviated as N, n=1), adult ( abbreviated as Ad, n=2), aging ( abbreviated as Ag, n=6), and stroke-injured ( abbreviated as I, n=1) hippocampus, respectively. (**E**) Pseudotime reconstruction of the neurogenic lineage in the stroke-injured human hippocampus. Different colors represent different cell types. The arrow indicates the developmental direction. (**F**) Heatmap showing the expression profiles of differentially expressed genes (DEGs) in four clusters along the pseudotime. Representative DEGs and enriched GO terms of each cluster are shown (GO:BP, regeneration related GO terms, p<0.05). (**G**) The significantly up-regulated genes in neurogenic lineage upon injury compared with aging. (**H**) The GO term analysis of up-regulated genes in the neurogenic lineage upon injury compared with aging (GO:BP, proliferation and regeneration related GO terms, p<0.05).

The online version of this article includes the following source data and figure supplement(s) for figure 6:

**Source data 1.** Genes and enriched gene ontology (GO) terms of *Figure 6F and H* and *Figure 6*.

**Figure supplement 1.** Stroke injury induced hippocampal cell apoptosis, astrocyte reactivation, and neuronal damages.

**Figure supplement 2.** Initially defined primed neural stem cells (pNSCs) and active NSCs (aNSCs) from stroke-injured hippocampus contained reactive astrocytes and reactivated NSCs.

**Figure supplement 3.** Integration of our single-nucleus RNA-seq (snRNA-seq) dataset with other published data.

To reconstruct the injury-induced activation trajectory of qNSCs and explore their developmental potential, we excluded reactive astrocytes and included qNSC1, qNSC2, pNSC, aNSC, and NB as neurogenic lineage for further analysis. In agreement with previous studies of adult neurogenesis (*Dulken et al., 2017*; *Artegiani et al., 2017*), the trajectory originated from qNSC1, qNSC2, and progressed to pNSCs, and then to aNSCs or NBs (*Figure 6E*). Based on this trajectory, *HOPX* and *PAX6* were mainly expressed where qNSCs were located, then *VIM, CD44, TNC,* and *CHI3L1* reached their expression peaks in the middle of the trajectory where pNSCs located, followed by *SOX2, CKAP5, RANGAP1* genes in aNSCs and *STMN2* gene in NBs (*Figure 6—figure supplement 2D*). The trajectory and gene expression together support that qNSCs can be activated to become pNSCs and aNSCs. Since the patient did not live long after the stroke, we attempted to predict the developmental potential of NSC lineages by analyzing the gene expression cascade along with the pseudotime. According to the gene expression cascade, DEGs corresponding to four clusters were identified (*Figure 6F*). *TNC, SOX2, LPAR1,* and *CLU* were highly expressed at the root of the trajectory (cluster 1). Consistently, genes from the cluster 1 were related to canonical Wnt signaling pathway, tissue regeneration, stem cell proliferation, and neuronal stem cell population maintenance. Subsequently, genes from cluster 2 were enriched with the generation of neurons, dendrite, and glial cell development, such as *PTPRG, NRG3, LGI1,* and *SLC4A4*; and lastly, genes for neuronal function (e.g. *MAP2, SEMA5A, SYT1, SYN2, SYN3,* and *MYT1L*) and glial fate determination (*LAMP2, PLP1, MBP,* and *MOG*) became dominant at the end of the trajectories fate 1 and fate 2 (clusters 3 and 4) (*Figure 6F*). Accordingly, the enriched GOs of genes from cluster 4 (fate 1) were related to neurogenesis, neuron projection development, neurotransmitter secretion, and synapse organization; the enriched GOs of genes from cluster 3 (fate 2) were associated with glial cell differentiation and myelination (*Figure 6F* and *Figure 6—source data 1*). Together, our data indicate that stroke-induced injury triggers activation of qNSCs, which then generate pNSCs and aNSCs, the latter of which have the potential to give rise to either neurons or oligodendrocytes (*El Waly et al., 2018*; *Parras et al., 2004*; *Llorens-Bobadilla et al., 2015*; *Koutsoudaki et al., 2016*).

To understand relationships between regeneration and the hippocampal neurogenic lineage after stroke injury, we next explored genes involved in the activation of neurogenic lineages (qNSC1, qNSC2, pNSC, aNSC, NB). We hypothesized that genes up-regulated upon injury are likely responsible for driving NSC activation. Therefore, we compared the expression of neurogenic lineage genes between the aged and injured hippocampus (*Figure 6G and H*). Specific genes that were significantly increased in the injured hippocampus were related to regeneration, autophagy, proliferation, inflammation, and metabolism (*Figure 6G and H*), some of which functions have previously been demonstrated. In mice, *Tnc, Gpc6, Cryab,* and *Gbp2* were reported to promote neuron regeneration and synapse formation following stroke-induced injury (*Chen et al., 2021*; *Saglam et al., 2021*; *Chen et al., 2010*; *Ugalde et al., 2020*); *Vmp1, Chi3l1, Spp1, Vim,* and *Itgb1* associated with autophagy, proliferation, and regeneration (*Zhao et al., 2017*; *Nishimura et al., 2021*; *Sojan et al., 2022*; *Kong et al., 2018*); and *Cyr61, CD63, Actn4, Ell2,* and *Spocd1* demonstrated to promote proliferation (*Kong et al., 2018*; *Thines et al., 2022*; *Chen et al., 2022*; *Alexander et al., 2017*; *Liu et al., 2018*).

Furthermore, *IFI44L, RASGEF1B,* and *STAT1* are linked with both inflammation and metabolic functions (*Cooles et al., 2022*; *Leão et al., 2020*) and *ACTN2, GBE1,* and *NAMPT* only for metabolic functions (*Ebersole et al., 2018*; *Gasparrini et al., 2022*). Overall, the stroke-induced up-regulated molecular signatures capture a broad activation state and regeneration of the neurogenic lineage.

## Discussion

The existence of human adult hippocampal neurogenesis has been a topic of debate over the years. Sample rarity and technical limitations are barriers that prevent us from investigating the human postnatal hippocampus during aging and post injury. With the development of snRNA-seq technology, we are able to better understand the blueprint of hippocampal neurogenesis signatures in humans. By using snRNA-seq technology, two recent studies found no adult neurogenic trajectories in human brains (*Franjic et al., 2022*; *Ayhan et al., 2021*), while other two groups newly reported that noticeable amounts of NSCs and immature neurons were found in the adult and aged human hippocampi, supporting adult human neurogenesis capacity (*Zhou et al., 2022*; *Wang et al., 2022*). While accumulated publications support the existence of neurogenesis in the adult human hippocampus, the homeostasis and developmental potentials of NSCs under different contexts remain unclear. Here, we have revealed the heterogeneity and developmental trajectory of hippocampal NSCs, and captured its transcriptional molecular dynamics during postnatal development, aging, and injury, which the traditional immunostainings could not uncover based on the limited sensitivity and specificity. Specially, we identified NSCs with different refined transcriptional statuses, including qNSC, pNSC, and aNSC populations. Despite transcriptional similarity between qNSCs and astrocytes, we also distinguished qNSCs from astrocytes by using gene set score analysis.

The lack of specific markers has prevented the identification of neurogenic lineages in the human hippocampus for a long time. To fill this gap, we executed an integrated cross-species analysis combined with scHPF and Seurat analysis to identify specific markers for human neurogenesis. In the study, we observed that both well-known and recently reported immature GC markers (*Zhou et al., 2022*; *Wang et al., 2022*; *Franjic et al., 2022*; *Hao et al., 2022*), such as DCX, PROX1, and STMN2, are widely expressed in human GABA-INs, which is consistent with Franjic's observation (*Franjic et al., 2022*).

It suggests the risk of interneuron contamination when using these markers to identify immature GCs. We further identified new specific NB markers by excluding genes expressed in human GABA-INs, such as CALM3, NEUROD2, NRGN, and NGN1. Thus, our findings extend our knowledge about the maker specificity of human adult hippocampal neurogenesis.

In agreement with recent studies, we also found that ETNPPL as an NSC marker (*Wang et al., 2022*) was highly expressed in our identified qNSC1/2 (*Figure 6—figure supplement 3A*), and NBs with the positivity of STMN1/2 (*Wang et al., 2022*) were maintained in the adult hippocampus (*Figure 3B and C*). In contrast, we did not find a comparable number of pNSCs, aNSCs, and imGCs as reported in the aged group, but detected reactivated NSCs in the injured hippocampus. To explore the cause of the discrepancies, we examined the published human specimens' information from different studies which reported the existence of NBs in the aged hippocampi (*Zhou et al., 2022*). When we integrated Zhou's snRNA-seq dataset of 14 aged donors (from 60 to 92 years of age) with our snRNA-seq dataset, we did not detect evident pNSC, aNSC, or NB populations in their 14 aged donors (*Figure 6—figure supplement 3B*). To rule out the possibility of missing cell clusters caused by analysis of Zhou's data, we examined the expression of pNSC/aNSC markers (e.g. VIM, TNC) and NB markers (e.g. STMN1 and NRGN), and they were not enriched in putative pNSC/aNSC/NB clusters, neither in other clusters (*Figure 6—figure supplement 3C and D*). However, EdU$^+$PROX1$^+$ newborn GCs were observed in surgically resected young and adult human hippocampi from patients diagnosed with epilepsy, temporal lobe lesions, or suspected low-grade glioma after in vitro culture (*Zhou et al., 2022*). One possibility is that these newborn GCs were originated from the injury-induced activated NSCs caused by the process of hippocampus sectioning or in vitro culture. In addition, we noticed that two aged donors diagnosed with rectal cancer (M67Y) and uterine tumor (F52Y) in Wang's study still maintained neurogenesis (*Wang et al., 2022*). Given recent evidence of crosstalk between cancer and neurogenesis (*Silverman et al., 2021*; *Mauffrey et al., 2019*), we suggest that cancer might provoke neurogenesis-like status in the adult human brain. Besides, Terreros-Roncal's work showed that amyotrophic lateral sclerosis, Huntington and Parkinson's disease could increase hippocampal neurogenesis

(*Terreros-Roncal et al., 2021*). Taking these data together, adult hippocampal neurogenesis is more easily to be detected in cases with neurological diseases, cancer, and injuries (*Terreros-Roncal et al., 2021*; *Zhou et al., 2022*; *Wang et al., 2022*). Therefore, the discrepancies among studies might be caused by health state differences across hippocampi, which subsequently lead to different degrees of hippocampal neurogenesis.

We constructed a developmental trajectory of NSCs in the neonatal hippocampus. Based on the trajectory and immunostaining analysis, we first deciphered transcriptional cascades of neurogenic lineages along with human hippocampal neurogenesis, and identified feature genes and TFs for each cell type. Combining the analysis of NSC properties and dynamics in neonatal, adult, aging, and injured human hippocampus, our results supported the process of NSCs from active to quiescent status during aging and their reactivation under injury. In our study, we detected NBs in the adult human hippocampus and active radial glial-like stem cells in the injured hippocampus by both immunostaining and snRNA-seq. The existence of NBs but not aNSCs in the adult hippocampus indicated a long maturation period of NBs in humans, in agreement with previous reports that the maturation period of NBs is longer in primates than in rodents (*Ngwenya et al., 2015*; *Seki, 2020*). Although a very rare number of NBs were captured by snRNA-seq, their presence was not validated by immunostaining. Because the donor died 2 days after the stroke, we surmise that there was not sufficient time for injury-induced aNSCs to fully differentiate into NBs. However, the obviously up-regulated neuronal and glial genes in the aNSC lineage in the injured hippocampus imply that these cells have the potential to generate neurons and glial cells. In addition to analyzing our own data, we also downloaded snRNA-seq data from *Zhou et al., 2022*, *Wang et al., 2022*, *Franjic et al., 2022*, and *Ayhan et al., 2021* for integrative analysis. While the dataset from Zhou et al. utilized machine learning and made it difficult to extract cell-type information for fitting with our own data, the datasets from the other three laboratories were successfully mapped onto our dataset. Based on the mapping analysis, AS2, qNSC, aNSC, and NB populations were identified with varying correlations in different datasets (*Figure 6—figure supplement 3E—G*). Combined our findings and the integrative analysis, the results together suggest that the reserved qNSCs in the adult human brain can be activated by stimuli such as injury or disease, and that their inherent neurogenesis capacity can be re-awakened by specific hippocampal microenvironments.

Taken together, our work deciphers the molecular heterogeneity and dynamics of human hippocampal NSCs under different contexts. This research provides valuable insights into the development, quiescence, and reactivation of human hippocampal NSCs, which may explain why adult hippocampal neurogenesis is generally difficult to observe in humans but can be detected in specific cases. However, we must acknowledge that the information about patients' health status and relevant lifestyle parameters is limited, and the number of patients in neonatal and stroke cases is very low (n=1). As a result, working with the current facts requires critical thinking and caution. We also realized that snRNA-seq has its limitations in distinguishing cells with very similar transcriptional signatures (such as qNSCs and astrocytes), and the function of the very rare number of NSCs or NBs that were captured by snRNA-seq without protein detection still needs to be further identified. Integrative analysis of epigenomic, proteomic, and metabolomic features of individual hippocampal cells and non-invasive lineage tracing in human brain will be more valued in the future.

## Materials and methods

### Human hippocampal sample collection

De-identified postnatal human hippocampus samples were obtained from the ZHONG-ZHI-YI-GU Research Institute. We recruited 10 donors from neonatal day 4 to 68 years of age (neonatal [postnatal 4 days], n=1; adult [31y, 32y], n=2; aging [50y, 56y, 60y, 64y-1, 64y-2, 68y], n=3; stroke injury [48y], n=1), consisting of 1 female and 9 males. Death reasons of these donors included: 1 congenital heart disease (postnatal day 4), 1 cerebral infarction (31y), 1 traumatic death (motor vehicle accident) (32y), 1 hypoxic-ischemic encephalopathy (stroke, 48y), 1 hypertension (50y), 3 carcinomas of the lungs (56y, 60y, 64y-2), 1 multiple organ failure (64y-1), and 1 carcinoma of the urinary bladder (68y) (*Figure 1—source data 1*). We dissected and collected the pair of hippocampi from the donors with a short post-mortem interval (about 3–4 hr). For individuals, the left hippocampus was used for snRNA-seq analysis; the right hippocampus was fixed for immunohistochemistry analysis. Given the differences

between the rostral and caudal hippocampus (*Wu and Hen, 2014*), we used the anterior and mid hippocampus containing typical DG structures for snRNA-seq and immunostaining.

## Cynomolgus monkey hippocampal sample collection

Female cynomolgus monkey, in age of 3 month with body weights of 2.3 kg, was used in this study. The anterior and mid hippocampus containing typical DG structures was collected for immunostaining.

## Isolation and purification of nuclei from adult human hippocampal tissues

The cell nuclei were isolated from frozen hippocampus according to the 10x Genomics nuclei isolation protocol for adult brain tissue with minor modifications (https://support.10xgenomics.com/single-cell-gene). Briefly, frozen hippocampus tissues with DG structures were minced with surgical scissors on ice. The minced tissues were transferred into a tube with Hibernate A (Gibco, PN-A1247501)/B27/GlutaMAX (*Dulken et al., 2017*) medium for equilibration. After the tissue was settled at the bottom of the tube, extra HEB was removed, leaving only enough medium to cover the tissue. Chilled lysis buffer (10 mM Tris-HCl, 10 mM NaCl, 3 mM MgCl$_2$, and 0.1% Nonidet P40 Substitute [Sigma-Aldrich, PN-74385]) was added to the tissue and the tube was incubated on ice for 15 min with gentle shaking during the incubation. Then tissues with lysis buffer were triturated with a Pasteur pipette for 10–15 passes to obtain a single-nuclei suspension. A 30 µm MACS SmartStrainer (Miltenyi Biotec, PN-130-098-458) was used to remove cell debris and large clumps. After centrifuging the nuclei at 500×*g* for 5 min at 4°C, Nuclei Wash and Resuspension Buffer (1× PBS with 1.0% bovine serum albumin [BSA] and 0.2 U/µl RNase inhibitor [Sigma-Aldrich, PN-3335399001]) was added and gently pipetted for 8–10 times. After two times washing, Myelin Removal Beads II slurry (Miltenyi Biotec, PN-130-096-733) was added to the nuclei pellet. After resuspension and wash, the LS column and magnetic separation were applied to remove the myelin. The cleaned nuclei pellet was resuspended for density gradient centrifugation with a sucrose cushion. After centrifugation, 700–1000 nuclei/µl was prepared for the following 10x Genomics Chromium capture and library construction protocol.

## snRNA library preparation for high-throughput sequencing

snRNA-seq libraries were generated by using Chromium Single Cell 3′ Reagent Kits v3, including three main steps: (1) Gel Bead-In-Emulsion (GEM) Generation and Barcoding; (2) Post GEM-RT Cleanup and cDNA Amplification; (3) 3′ Gene Expression Library Construction. Briefly, GEMs are generated by combining barcoded Single Cell 3′ v3 Gel Beads, a Master Mix containing cells, and Partitioning Oil onto Chromium Chip B. 5420–18,832 nuclei were captured per channel. To achieve single-nucleus resolution, nuclei were delivered at a limiting dilution. Immediately following GEMs generation, the Gel Beads were dissolved, primers containing an Illumina R1 sequence, a 16 bp 10x Barcode, a 10 bp randomer, and a poly-dT primer sequence were released and mixed with cell lysate and Master Mix. After incubation of the GEMs, barcoded, full-length cDNA from poly-adenylated mRNA was generated. Barcoded, full-length cDNA was amplified via PCR to generate sufficient mass for library construction. Prior to library construction, enzymatic fragmentation and size selection were used to optimize the cDNA amplicon size. P5 primer, P7 primer, sample index sequence, and TruSeq Read 2 (read 2 primer sequence) were added via end repair, A-tailing, adaptor ligation, and PCR. The final libraries containing the P5 and P7 primers were generated by Illumina bridge amplification. Sample index sequence was incorporated as the i7 index read. TruSeq Read 1 and TruSeq Read 2 were used in paired-end sequencing (http://10xgenomics.com). Finally, the library was sequenced as 150 bp paired-end reads by using the Illumina Nova6000.

## Filtering and normalization

The Cell Ranger Single-Cell Software Suit (3.0.2) (http://10xgenomics.com) (*Zheng et al., 2017*) was used to perform quality control and read counting of ensemble genes with default parameters (3.0.2) by mapping to the GRCh38 pre-mRNA reference genome. Only confidently mapped reads with valid barcodes and unique molecular identifiers were used to generate the gene-barcode matrix. We excluded poor quality cells after the gene-cell data matrix was generated by Cell Ranger software by using the Seurat package (4.0.3) (*Butler et al., 2018*; *Stuart et al., 2019*). Only nuclei that expressed more than 200 genes and fewer than 5000–8600 (depending on the peak of enrichment genes) genes

were considered. Cells with less than 200 genes or more than 8600 genes (likely cell debris and doublets) were removed. We also removed cells with more than 20% of the transcripts generated from mitochondrial genes. The co-isolation of mitochondria during the nucleus isolation process is likely due to their association with ER. This is consistent with reports from other groups where mitochondrial DNA was detected in snRNA-seq. In total, 33,538 genes across 92,966 single nuclei remained for subsequent analysis (postnatal day 4 remained 17,707 nuclei, 31y remained 12,406 nuclei, 32y remained 11,804 nuclei, 48y remained 15,398 nuclei, 50y remained 5543 nuclei, 56y remained 4665 nuclei, 60y remained 7597 nuclei, 64y-1 remained 5239 nuclei, 64y-2 remained 6309 nuclei, 68y remained 6298 nuclei).

## Single-cell clustering and visualization

We used the NormalizeData and FindVariableFeatures functions implemented in Seurat v3, performed standard preprocessing (log-normalization), and identified the top 2000 variable features for each individual dataset. We then identified integration anchors using the FindIntegrationAnchors function (*Satija et al., 2015*).We used default parameters and dimension 20 to find anchors. We then passed these anchors to the IntegrateData function to generate integrated Seurat object. To visualize the data, we used UMAP to project cells in 2D and 3D space based on the aligned canonical correlation analysis. Aligned canonical correlation vectors (1:20) were used to identify clusters using a shared nearest neighborhood modularity optimization algorithm.

## Identification of cell types based on DEGs

Using graph-based clustering, we divided cells into 35 clusters using the FindClusters function in Seurat with resolution 1 (*Butler et al., 2018*). We identified 16 cell types including two unknown populations. The identified cell types are: astrocytes and qNSC (*GFAP, HES1, NOTCH2*), pNSCs-qNSCs (*HOPX, VIM*), aNSCs (*CCND2, SOX2*), NB (*DCX, MYT1L*), GC (*SYT1, SV2B*), interneuron (*SST, CCK*), oligodendrocyte (*MOG*), microglia (*CSF1R*), pyramidal neurons (*PNN*), endothelial cells (*VWF*), oligodendrocyte precursor cell (*OLIG1, SOX10*), Reelin-expressing Cajal-Retzius cells (*RELN*), pericytes, and adult astrocyte (*S100B, ALDH1L1*). The DEGs of each cluster were identified using the FindAllMarkers function (thresh.use=0.25, test.use = 'wilcox') with the Seurat R package (6). We used the Wilcoxon rank-sum test (default), and genes with average expression difference >0.5 natural log and p<0.05 were selected as marker genes. Enriched GO terms of marker genes were identified using enricher function with the clusterProfiler package. Hierarchical clustering and heatmap generation were performed for single cells based on log-normalized expression values of marker genes curated from literature or identified as highly DEGs. Heatmaps were generated using the Heatmap function from the Complexheatmap v2.8.0 R package. To visualize the expression of individual genes, cells were grouped into different types determined by analysis with Seurat.

## Cell cycle analysis

In the cell cycle analysis, we applied a cell cycle-related gene set with 49 genes that are highly expressed in aNSCs than in other NSCs (astrocyte-qNSC, pNSC, and NB) during S and G2/M phase. UMAP plot of 92,966 single-nucleus transcriptomes with points colored by putative cell cycle phase (G0/G1, G2/M, or S) using the CellCycleScoring function in Seurat (*Macosko et al., 2015*; *Tirosh et al., 2016*).

## Gene set score analysis

Gene set scores (*Figure 2E and F*) were calculated by Seurat (AddModuleScore) according to previously defined RGL and reactive astrocyte gene sets (*Zamanian et al., 2012*; *Liddelow et al., 2017*; *Clarke et al., 2018*; *Hochgerner et al., 2018*; *Zhong et al., 2020*; *Franjic et al., 2022*) as control feature sets. These reference raw datasets are available in the NCBI Gene Expression Omnibus (GEO) repository, accession number:

GSE35338, GSE95753, GSE131258, GSE186538. Briefly, we calculated the average expression of each cell cluster on the single-cell level, subtracted by the aggregated expression of control feature sets. All analyzed features are binned based on averaged expression, and the control features are randomly selected from each bin.

## Pseudotime analysis of the neurogenic lineage in neonatal and stroke-injured hippocampal cells

The Monocle 2R package (v2.20.0) (*Qiu et al., 2017a*; *Trapnell et al., 2014*) were applied to construct single-cell pseudotime trajectories (*Qiu et al., 2017a*; *Qiu et al., 2017b*; *Trapnell et al., 2014*) to discover developmental transitions. Cells in Seurat clusters were inferred to be the corresponding locations of the neurogenesis differentiation axis. The pNSC or qNSC1 are at the beginning of pseudo-time in the first round of 'order Cells'. Dispersed genes used for pseudotime ordering were calculated by the 'estimateDispersions' function. 'DDR Tree' was applied to reduce dimensional space and the minimum spanning tree on cells was plotted by using the visualization function 'plot_cell_trajectory' for Monocle 2. Monocle function: reduceDimension(mycds, max_components = 2, method = 'DDR Tree').

## Expression heatmap of highly dynamically expressed genes along the pseudotime

pNSC generated two branches, GC subtypes GC1 and GC2, in neonatal 4 days trajectories. These branches will be characterized by distinct gene expression programs. Branched expression analysis modeling aims to find all genes that differ between the branches which contain four gene clusters in neonatal 4 days. Differentiation-related DEGs were obtained with a cutoff of q-value $<1 \times 10^{-4}$, and contained four gene clusters. In addition, the 'differentialGeneTest' function in Monocle 2R package was used to find all genes that differ between trajectory cell types (qNSC1, qNSC2, pNSC, aNSC) in stroke injury hippocampus.

## Comparison of DEGs in neurogenic lineage across aging process and injury condition

We obtained significantly up-regulated and down-regulated genes in aged hippocampal neurogenic lineages by comparing them with those in neonatal neurogenic lineages. Subsequently, we visualized these DEGs in neonatal, middle-aged, and aged neurogenic lineages by violin plot and heatmap. To explore the DEGs under the stroke injury condition, we compared gene expressions of neurogenic lineages between aged and stroke-injured hippocampus. We visualized these DEGs from neurogenic lineages in neonatal, adult, aging, and stroke injury hippocampus by bubble chart to show their differential expression.

## Prediction of biological functions by GO term analysis

We enriched DEGs in neurogenic lineages during aging and under stroke injury conditions by GO term analysis. GO analysis was performed by the clusterProfiler package.

### scHPF and Seurat analysis (FindAllMarkers)

To identify new marker gene signatures associated with neurogenic lineages including qNSC1, qNSC2, pNSC, aNSC, and NB in neonatal 4 days, we factorized the data with scHPF (*Levitin et al., 2019*) and Seurat analysis (FindAllMarkers) from different factors onto the neurogenic lineage. To select the optimal number of factors, first, we ran scHPF for different numbers of factors, K (from 2 to 20, interval 1).Optimal effect was obtained when it had a value of 7. We picked the model with K=7 and presented top 10 marker genes of scHPF analysis (*Figure 3A*). Meanwhile, we presented top 15 gene markers of FindAllMarkers function of Seurat analysis (*Figure 3B*).

### Multimodal reference mapping

The 'multimodal reference mapping' introduces the process of mapping query datasets to annotated references in Seurat v4. By using Seurat v4 and SeuratDisk package, we mapped Wang et al. (*Cell Research*, 2022a), Franjic et al. (*Neuron*, 2022), and Ayhan et al. (*Neuron*, 2021) human hippocampal snRNA datasets to our human hippocampal datasets. These reference raw datasets are available in the NCBI GEO repository, accession number: GSE163737, GSE186538, GSE160189. First, annotate each query cell based on a set of reference-defined cell states. Second, project each query cell onto our previously computed UMAP visualization.

## Immunostaining of human and monkey hippocampal tissues

The hippocampus from the right side of the human brain with a short post-mortem interval was dissected. Monkey is deeply sedated with isoflurane and then euthanized with an overdose of pentobarbital. The monkey brain was removed from the skull, and the hippocampus was obtained. The human and monkey hippocampal tissue fixed with 4% paraformaldehyde (PFA) for up to 24 hr and cryoprotected in 30% sucrose at 4°C until completely sink to the bottom. The tissue samples were frozen in OCT (Tissue-Tek) on dry ice and sectioned at 10 μm on a cryostat microtome (Leica CM1950). Tissue slides sectioned from the anterior of the hippocampus containing typical DG structures were first incubated in blocking and permeation solution with 2% Triton X-100 (Sigma) for 2 hr. Next, the sections were treated with a VECTOR TrueView autofluorescence quenching kit (Vectorlabs, PN-SP-8500-15) to reduce the innate autofluorescence of the human tissue, washed with 3×15 min PBS (pH 7.6), and then incubated in 3% BSA for 1 hr at room temperature (RT). Subsequently, sections were incubated overnight at 4°C with the following primary antibodies: anti-Nestin (rabbit, 1:500, Millpore, PN-MAB5326) and anti-ki67 (mouse, 1:500, R&D Systems, PN-AF7649); anti-CHI3L1 (rabbit, 1:200, Novus Biologicals, PN-NBPI-57913; rabbit, 1:100, Proteintech, 12036-1-AP); anti-Vimentin (rabbit, 1:300, Abcam, PN-ab137321); anti-Vimentin (mouse, 1:800, eBioscience, PN-14-9897); anti-HOPX (rabbit, 1:500, Sigma, PN-HPA030180); anti-PSA-NCAM (mouse, 1:500, Millipore, PN-MAB5324); SST, mouse (sc-55565, Santa Cruz Biotechnology). After overnight incubation, tissue sections were washed with PBS for 3×15 min, and then incubated with secondary antibodies at RT for 2 hr: Alexa Fluor 488 AffiniPure donkey anti-rabbit IgG(H+L) (1:500, Jackson Immunoresearch, PN-712-545-152), Alexa Fluor 647 AffiniPure donkey anti-rabbit IgG(H+L) (1:500, Jackson Immunoresearch, PN-715-605-150). DAPI staining (Sigma, PN-32670-5mg-F), Donkey anti-Rabbit IgG (H+L) Highly Cross-Adsorbed Secondary Antibody, Alexa Fluor 555 (1:500, Thermo Fisher, PN- A-31572) was performed and sections were washed with 1× PBS for 3×15 min. After washing, sections were mounted and dried, ready for microscope observation.

## TUNEL assay

Tissue sections were analyzed for DNA fragmentation using a TUNEL-based method (BOSTER, PN-MK1012). Briefly, sections were first permeabilized in 0.02% Triton X-100 overnight. To label damaged nuclei, 20 μl of the TUNEL reaction mixture (Labeling buffer, TdT, BIO-d-UTP) was added to each sample and kept at 37°C in a humidified chamber for 120 min. Sections were washed with PBS for 2 min and blocked with 50 μl blocking reagent at RT for 30 min. Then SABC buffer and DAPI were added following the protocol of BOSTER TUNEL kit for the detection of apoptotic cells.

## Human hippocampal tissues and ethics statement

This work was approved by the ZHONG-ZHI-YI-GU Research Institute of Human Research Protection (ZZYG-YC2019-003). All donated tissues in this study were from dead patients. Tissue was collected following the guidelines recommended by the Ethical Review of Biomedical Research Involving People for tissue donation. Hippocampus tissue samples were collected after the donor patients (or family members) signed an informed consent document that was in strict observance of the legal and institutional ethics at ZHONG-ZHI-YI-GU Research Institute. All hippocampal samples used in these studies had not been involved in any other procedures. All the protocols followed the Interim Measures for the Administration of Human Genetic Resources, administered by the Ministry of Science and Technology of China.

## Cynomolgus monkey hippocampal tissues and ethics statement

Animal ethics statement: Female cynomolgus monkeys, in age of 3 month with body weights of 2.3 kg, were used in this study. All animals were housed at Kunming University of Science and Technology (KUST), and individually bred in an American standard cage at a light/dark cycle of 12 hr/12 hr. Reference Number of the Research Ethics Committee, Kunming University of Science and Technology: KUST202301005. All animal procedures were approved in advance by the Institutional Animal Care and Use Committee of Kunming University of Science and Technology and were performed in accordance with the Association for Assessment and Accreditation of Laboratory Animal Care International for the ethical treatment of primates.

## Acknowledgements

This work was supported by the National Key Research and Development Program of China (2022YFA1103100), the Key Research Project of Science and Technology of Yunnan (YNWR-YLXZ-2020-015), the National Natural Science Foundation of China (NSFC) (32070864 and 32160153), STI2030-Major Projects (2022ZD0207700), Xingdian Talent Support Program (KKRD202273100), Major Basic Research Project of Science and Technology of Yunnan (202001BC070001 and 202102AA100053), and the Natural Science Foundation of Yunnan Province (202105AD160008, 202101AT070287, and 202207AA110003).

## Additional information

### Funding

| Funder | Grant reference number | Author |
| --- | --- | --- |
| National Key Research and Development Program of China | 2022YFA1103100 | Tianqing Li |
| The Key Research Project of Science and Technology of Yunnan | YNWR-YLXZ-2020-015 | Tianqing Li |
| National Natural Science Foundation of China | 32070864 and 32160153 | Shaoxing Dai Runrui Zhang |
| STI2030-Major Projects | 2022ZD0207700 | Runrui Zhang |
| Xingdian Talent Support Program | KKRD202273100 | Runrui Zhang |
| Major Basic Research Project of Science and Technology of Yunnan | 202001BC070001 and 202102AA100053 | Weizhi Ji |
| Natural Science Foundation of Yunnan Province | 202105AD160008 | Shaoxing Dai Runrui Zhang |
| Natural Science Foundation of Yunnan Province | 202207AA110003 | Shaoxing Dai Runrui Zhang |
| Natural Science Foundation of Yunnan Province | 202101AT070287 | Shaoxing Dai Runrui Zhang |

The funders had no role in study design, data collection and interpretation, or the decision to submit the work for publication.

### Author contributions

Junjun Yao, Formal analysis, Validation, Investigation, Visualization, Methodology, Writing - original draft, Writing - review and editing; Shaoxing Dai, Data curation, Software, Formal analysis, Supervision, Funding acquisition, Methodology; Ran Zhu, Data curation, Software, Formal analysis, Validation, Investigation, Visualization, Methodology; Ju Tan, Jiansen Sun, Resources, Investigation, Methodology; Qiancheng Zhao, Validation, Investigation, Visualization, Methodology; Yu Yin, Software, Methodology; Xuewei Du, Nan Li, Jun Li, Methodology; Longjiao Ge, Validation, Methodology; Jianhua Xu, Resources, Investigation; Chunli Hou, Resources, Methodology; Weizhi Ji, Funding acquisition; Chuhong Zhu, Resources, Supervision, Funding acquisition; Runrui Zhang, Formal analysis, Supervision, Funding acquisition, Methodology, Writing - original draft, Project administration, Writing - review and editing; Tianqing Li, Conceptualization, Supervision, Funding acquisition, Project administration, Writing - review and editing

### Author ORCIDs

Junjun Yao http://orcid.org/0000-0002-5190-4611

Ran Zhu http://orcid.org/0000-0003-0874-9369
Nan Li http://orcid.org/0000-0003-2111-4835
Runrui Zhang http://orcid.org/0000-0003-2871-0859
Tianqing Li http://orcid.org/0000-0001-7453-0184

## Ethics

This work was approved by the ZHONG-ZHI-YI-GU Research Institute of Human Research Protection (ZZYG-YC2019-003). All donated tissues in this study were from dead patients. Tissue was collected following the guidelines recommended by the Ethical Review of Biomedical Research Involving People for tissue donation. Hippocampus tissue samples were collected after the donor patients (or family members) signed an informed consent document that was in strict observance of the legal and institutional ethics at ZHONG-ZHI-YI-GU Research Institute. All hippocampal samples used in these studies had not been involved in any other procedures. All the protocols followed the Interim Measures for the Administration of Human Genetic Resources, administered by the Ministry of Science and Technology of China.

Female cynomolgus monkeys, in age of 3 month with bodyweights of 2.3 kg, were used in this study. All animals were housed at Kunming University of Science and Technology (KUST), and individually bred in an American standard cage at a light/dark cycle of 12 hours/12 hours. Reference Number of the Research Ethics Committee, Kunming University of Science and Technology: KUST202301005. All animal procedures were approved in advance by the Institutional Animal Care and Use Committee of Kunming University of Science and Technology and were performed in accordance with the Association for Assessment and Accreditation of Laboratory Animal Care International for the ethical treatment of primates.

Reviewer #1 (Public review): https://doi.org/10.7554/eLife.89507.4.sa1
Reviewer #2 (Public review): https://doi.org/10.7554/eLife.89507.4.sa2
Author response https://doi.org/10.7554/eLife.89507.4.sa3

# Additional files

## Supplementary files
• MDAR checklist

## Data availability

The accession numbers for the raw snRNA-seq data reported in this paper in Genome Sequence Archive (GSA): HRA003049. Specimen information and sequencing statistics are described in *Figure 1—source data 1*. The code used to perform analyses in this paper is available on GitHub at https://github.com/BigreyR/snRNA-seq_hippocampus (copy archived at *Bigrey, 2024*).

The following dataset was generated:

| Author(s) | Year | Dataset title | Dataset URL | Database and Identifier |
|---|---|---|---|---|
| Yao J, Dai S, Zhu R, Tan J | 2024 | Decoding development, aging and activation of neurogenic lineage in human postnatal hippocampus | https://bigd.big.ac.cn/gsa-human/browse/HRA003049 | bigd, HRA003049 |

The following previously published datasets were used:

| Author(s) | Year | Dataset title | Dataset URL | Database and Identifier |
|---|---|---|---|---|
| Wang W, Wang M, Yang M, Zeng B, Qiu W | 2022 | Adult hippocampal neurogenesis in aged macaques and humans | https://www.ncbi.nlm.nih.gov/geo/query/acc.cgi?acc=GSE163737 | NCBI Gene Expression Omnibus, GSE163737 |

*Continued on next page*

*Continued*

| Author(s) | Year | Dataset title | Dataset URL | Database and Identifier |
|---|---|---|---|---|
| Franjic D, Skarica M, Ma S | 2022 | Transcriptomic Taxonomy and Neurogenic Trajectories of Adult Human, Macaque and Pig Hippocampal and Entorhinal Cells | https://www.ncbi.nlm.nih.gov/geo/query/acc.cgi?acc=GSE186538 | NCBI Gene Expression Omnibus, GSE186538 |
| Ayhan F, Kulkarni A, Berto S, Sivaprakasam K | 2021 | Resolving cellular and molecular diversity along the hippocampal anterior-to-posterior axis in humans | https://www.ncbi.nlm.nih.gov/geo/query/acc.cgi?acc=GSE160189 | NCBI Gene Expression Omnibus, GSE160189 |
| Zhou Y, Su Y, Li S, Kennedy BC | 2022 | Dissecting the transcriptome landscape of the human hippocampus | https://www.ncbi.nlm.nih.gov/geo/query/acc.cgi?acc=GSE185553 | NCBI Gene Expression Omnibus, GSE185553 |
| Zhou Y, Su Y, Li S, Kennedy BC | 2022 | Molecular landscape of immature neurons in the human hippocampus across the lifespan | https://www.ncbi.nlm.nih.gov/geo/query/acc.cgi?acc=GSE185277 | NCBI Gene Expression Omnibus, GSE185277 |
| Zhou Y, Su Y, Li S, Kennedy BC | 2022 | Molecular landscape of immature neurons in the human hippocampus in Alzheimer's disease | https://www.ncbi.nlm.nih.gov/geo/query/acc.cgi?acc=GSE198323 | NCBI Gene Expression Omnibus, GSE198323 |

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

# Appendix 1

### Appendix 1—key resources table

| Reagent type (species) or resource | Designation | Source or reference | Identifiers | Additional information |
|---|---|---|---|---|
| Biological sample (*Homo sapiens*) | Neonatal hippocampal tissue (4 days, male) | ZHONG-ZHI-YI-GU Research Institute | | Freshly isolated from donors' brain PMI 3–5 hr |
| Biological sample (*Homo sapiens*) | Adult hippocampal tissue (31 years of age, male) | ZHONG-ZHI-YI-GU Research Institute | | Freshly isolated from donors' brain PMI 3–5 hr |
| Biological sample (*Homo sapiens*) | Adult hippocampal tissue (32 years of age, male) | ZHONG-ZHI-YI-GU Research Institute | | Freshly isolated from donors' brain PMI 3–5 hr |
| Biological sample (*Homo sapiens*) | Stroke hippocampal tissue (48 years of age, male) | ZHONG-ZHI-YI-GU Research Institute | | Freshly isolated from donors' brain PMI 3–5 hr |
| Biological sample (*Homo sapiens*) | Aged hippocampal tissue (50 years of age, male) | ZHONG-ZHI-YI-GU Research Institute | | Freshly isolated from donors' brain PMI 3–5 hr |
| Biological sample (*Homo sapiens*) | Aged hippocampal tissue (56 years of age, male) | ZHONG-ZHI-YI-GU Research Institute | | Freshly isolated from donors' brain PMI 3–5 hr |
| Biological sample (*Homo sapiens*) | Aged hippocampal tissue (60 years of age, male) | ZHONG-ZHI-YI-GU Research Institute | | Freshly isolated from donors' brain PMI 3–5 hr |
| Biological sample (*Homo sapiens*) | Aged hippocampal tissue (64 years of age, female) | ZHONG-ZHI-YI-GU Research Institute | | Freshly isolated from donors' brain PMI 3–5 hr |
| Biological sample (*Homo sapiens*) | Aged hippocampal tissue (64 years of age, male) | ZHONG-ZHI-YI-GU Research Institute | | Freshly isolated from donors' brain PMI 3–5 hr |
| Biological sample (*Homo sapiens*) | Aged hippocampal tissue (68 years of age, male) | ZHONG-ZHI-YI-GU Research Institute | | Freshly isolated from donors' brain PMI 3–5 hr |
| Biological sample (*Macaca sapiens*) | Cynomolgus monkey hippocampal tissues (3 months, male) | Kunming University of Science and Technology (KUST) | | Freshly isolated from monkey's brain |
| Chemical compound, drug | Hibernate A | Gibco | Cat# A1247501 | |
| Chemical compound, drug | B27 | Thermo | Cat# 17504044; CAS: 145567-32-4 | |
| Chemical compound, drug | GlutaMAX | Invitrogen | Cat# 35050-061 | |
| Chemical compound, drug | Tris-HCl | Beyotime | Cat# ST768; CAS: 92451-00-8 | |

*Appendix 1 Continued on next page*

*Appendix 1 Continued*

| Reagent type (species) or resource | Designation | Source or reference | Identifiers | Additional information |
|---|---|---|---|---|
| Chemical compound, drug | NaCl | Aladdin | Cat# C111533; CAS: 14762-51-7 | |
| Chemical compound, drug | MgCl$_2$ | Aladdin | Cat#M113692; CAS: 7786-30-3 | |
| Chemical compound, drug | 0.1% Nonidet P40 | Sigma-Aldrich | Cat# 74385 | |
| Chemical compound, drug | 30 µm MACS SmartStrainer | Miltenyi Biotec | Cat# 130-098-458 | |
| Chemical compound, drug | PBS | Solarbio | Cat# P1010 | |
| Chemical compound, drug | BSA | BOSTER | Cat# AR0189 | |
| Chemical compound, drug | RNase inhibitor | Sigma-Aldrich | Cat# 3335399001 | |
| Chemical compound, drug | Myelin Removal Beads II slurry | Miltenyi Biotec | Cat# 130-096-733 | |
| Chemical compound, drug | PFA | Sigma | Cat# 158127; CAS: 30525-89-4 | |
| Chemical compound, drug | Triton X-100 | BioFroxx | Cat# 1139ML100 | |
| Chemical compound, drug | DAPI staining | Sigma | Cat# 32670-5mg-F | 1:1000 |
| Commercial assay or kit | TUNEL-based method | BOSTER | Cat# MK1012 | |
| Commercial assay or kit | VECTOR TrueView autofluorescence quenching kit | Vectorlabs | Cat# SP-8500-15 | |
| Antibody | Anti-Nestin (rabbit monoclonal) | Millipore | Cat# MAB5326; RRID: AB_2251134 | 1:800 |
| Antibody | Anti-ki67 (mouse monoclonal) | R&D Systems | Cat# AF7649; RRID: AB_2687500 | 1:500 |
| Antibody | Anti-CHI3L1 (rabbit monoclonal) | Proteintech | Cat# 12036-1-AP; RRID: AB_2877819 | 1:500 |
| Antibody | Anti-Vimentin (rabbit monoclonal) | Abcam | Cat# ab137321; RRID: AB_2921312 | 1:800 |
| Antibody | Anti-Vimentin (mouse monoclonal) | Thermo Fisher Scientific | Cat# 14-9897-37; RRID: AB_2865507 | 1:500 |
| Antibody | Anti-HOPX (rabbit monoclonal) | Sigma | Cat# HPA030180; RRID: AB_10603770 | 1:1000 |
| Antibody | Anti-PSA-NCAM (mouse monoclonal) | Millipore | Cat# MAB5324; RRID: AB_95211 | 1:500 |
| Antibody | Anti-SST (mouse monoclonal) | Santa Cruz Biotechnology | Cat# sc-55565; RRID: AB_831726 | 1:500 |
| Antibody | Alexa Fluor 488 AffiniPure donkey anti-rabbit (secondary antibody) | Jackson Immunoresearch | Cat# 712-545-150; RRID: AB_2340683 | 1:500 |

*Appendix 1 Continued*

| Reagent type (species) or resource | Designation | Source or reference | Identifiers | Additional information |
|---|---|---|---|---|
| Antibody | Alexa Fluor 647 AffiniPure donkey anti-rabbit (secondary antibody) | Jackson Immunoresearch | Cat# 715-605-150; RRID: AB_2340862 | 1:500 |
| Antibody | Alexa Fluor 555 donkey anti-rabbit (secondary antibody) | Thermo Fisher | Cat# A-31572; RRID: AB_162543 | 1:500 |
| Software, algorithm | Seurat package (4.0.3) | *Hao et al., 2022* | https://satijalab.org/seurat/ | |
| Software, algorithm | Complexheatmap v2.8.0 R package | *Gu, 2022* | https://github.com/jokergoo/ComplexHeatmap | |
| Software, algorithm | Monocle 2R package (v2.20.0) | *Trapnell et al., 2014* | https://cole-trapnell-lab.github.io/monocle-release/ | |
| Software, algorithm | clusterProfiler package | *Yu et al., 2012* | https://github.com/JPingAMMS/clusterProfiler | |
| Software, algorithm | Single-cell hierarchical Poisson factorization (scHPF) | *Levitin et al., 2019* | https://github.com/simslab/scHPF | |

