## [Editor Report · eLife assessment]

Using state-of-the-art single-nucleus RNA sequencing, Yao et al. investigate the transcriptomic features of neural stem cells (NSCs) in the human hippocampus to address how they vary across different age groups and stroke conditions. The authors report alterations in NSC subtype proportions and gene expression profiles after stroke. Although the study is **valuable** and the analysis is comprehensive, the significance is restricted by well-acknowledged technical limitations leading to **incomplete** evidence supporting some main conclusions.

---

## [Referee Report · Reviewer #1 (Public review)]

In this manuscript, Yao et al. explored the transcriptomic characteristics of neural stem cells (NSCs) in the human hippocampus and their changes under different conditions using single-nucleus RNA sequencing (snRNA-seq). They generated single-nucleus transcriptomic profiles of human hippocampal cells from neonatal, adult, and aging individuals, as well as from stroke patients. They focused on the cell groups related to neurogenesis, such as neural stem cells and their progeny. They revealed genes enriched in different NSC states and performed trajectory analysis to trace the transitions among NSC states and towards astroglial and neuronal lineages in silico. They also examined how NSCs are affected by aging and injury using their datasets and found differences in NSC numbers and gene expression patterns across age groups and injury conditions. One major issue of the manuscript is questionable cell type identification. For example, more than 50% of the cells in the astroglial lineage clusters are NSCs, which is extremely high and inconsistent with classic histology studies.

While the authors have made efforts to address previous critics, major concerns have not been adequately addressed, including a very limited sample size and patient information. In addition, some analytical approaches are still questionable and the authors acknowledge some issues they cannot address. Therefore, while the topic is interesting, some results are preliminary and some conclusions are not fully supported by the data presented.

---

## [Referee Report · Reviewer #2 (Public review)]

In this manuscript, Yao et al. present a series of experiments aiming at generating a cellular atlas of the human hippocampus across aging, and how it may be affected by injury, in particular, stroke. Although the aim of the study is interesting and relevant for a larger audience, due to the ongoing controversy around the existence of adult hippocampal neurogenesis in humans, a number or technical weaknesses result in a poor support for many of the conclusions made from the results of these experiments.

In particular, a recent meta analysis of five previous studies applying similar techniques to human samples has identified different aspects of sample size as main determinants of the statistical power needed to make significant conclusions. Some of this aspects are the number of nuclei sequenced and subject stratification. These two aspects are of concern in Yao's study. First, the number of sequenced nuclei is lower than the calculated numbers of nuclei required for detecting rare cell types. However, Yao et al. report succeeding in detecting rare populations, including several types of neural stem cells in different proliferation states, which have been demonstrated to be extremely scarce by previous studies. It would be very interesting to read how the authors interpret these differences. Secondly, the number of donors included in some of the groups is extremely low (n=1) and the miscellaneous information provided about the donors is practically inexistent. As individual factors such as chronic conditions, medication, lifestyle parameters, etc... are considered determinant for the variability of adult hippocampal neurogenesis levels across individuals, this represents a series limitation of the current study. Overall, several technical weaknesses severely limit the relevance of this study and the ability of the authors to achieve their experimental aims.

After a first review round, the manuscript is still lacking a clear discussion of its several technical limitations, which will help the audience to grasp the relevance of the findings. In particular, detailed information about individual patients health status and relevant lifestyle parameters that may have affected it is lacking. The authors make the point themselves that the discrepancies among studies might be caused by health state differences across hippocampi, which subsequently lead to different degrees of hippocampal neurogenesis." So, even in the authors own interpretation this is a serious limitation to the manuscript, that however out of the authors control, impacts on the quality of their findings.

---

## [Author Response]

The following is the authors’ response to the previous reviews.

**Reviewer #1 (Public Review):**
In this manuscript, Yao et al. explored the transcriptomic characteristics of neural stem cells (NSCs) in the human hippocampus and their changes under different conditions using single-nucleus RNA sequencing (snRNA-seq). They generated single-nucleus transcriptomic profiles of human hippocampal cells from neonatal, adult, and aging individuals, as well as from stroke patients. They focused on the cell groups related to neurogenesis, such as neural stem cells and their progeny. They revealed genes enriched in different NSC states and performed trajectory analysis to trace the transitions among NSC states and towards astroglial and neuronal lineages in silico. They also examined how NSCs are affected by aging and injury using their datasets and found differences in NSC numbers and gene expression patterns across age groups and injury conditions. One major issue of the manuscript is questionable cell type identification. For example, more than 50% of the cells in the astroglial lineage clusters are NSCs, which is extremely high and inconsistent with classic histology studies.While the authors have made efforts to address previous critics, major concerns have not been adequately addressed, including a very limited sample size and with poor patient information. In addition, some analytical approaches are still questionable and the authors acknowledged that some they cannot address. Therefore, while the topic is interesting, some results are preliminary and some conclusions are not fully supported by the data presented.

We thank the reviewer for reevaluating our revised manuscript. We respect the reviewer’s comments and discuss the technical and conceptual limitations of this work. Here we provide the response to Reviewer #1 (Public Review) on these below.

Firstly, we appreciate the concerns raised by Reviewer 1 regarding the high proportion of NSCs within the astroglia lineage clusters. it is worth mentioning that distinguishing hippocampal qNSCs from astrocytes by transcription profiling poses a significant challenge in the field due to their high transcriptional similarity. From previous global UMAP analysis, AS1 (adult specific) can be separated from qNSCs, but AS2 (NSC-like astrocytes) cannot. Therefore, the data presented in Figure 2C to G aimed to further distinguish the qNSCs from AS2 by using gene set scores analysis. Based on different scores, we categorized qNSC/AS lineages into qNSC1, qNSC2 and AS2. Figure 2C presented the UMAP plot of qNSC/AS2 population from only neonatal sample. We apologize for not clarifying this in the figure legend. We have now clarified this information in the figure legend of Figure 2C. More importantly, we have added UMAP plots and quantifications for other groups in Figure 2-Supplement 2A and B, including adult, aging, and injure samples. This supplementary figure provides more complete information of the cell type composition and dynamic variations during aging and injury. Although the ratio of NSCs in the astroglia lineage clusters remains higher compared to classic histology studies, the trends indicate a reduction in qNSCs and an increase in astrocytes during aging and injury, which supports that cell type identification by using gene set score analysis is effective, although still not optimal. Combined methods to accurately distinguish between qNSCs and astrocytes are required in the future, and we also discuss this in the corresponding texts.

Secondly, we cannot adequately address the major concern regarding sample size raised by the reviewer due to the scarcity of stroke and neonatal human brain samples. We have collected additional details about the donors. Please refer to Figure 1-source data 1 for the updated information. Other information regarding the lifestyle parameters of these donors has not been sufficiently recorded by the hospital. Therefore, we cannot improve the patient information further.

Thirdly, regarding the questionable subpopulations of granule cells (GCs) that derive from neuroblasts in Figure 4A-4D, which are inconsistent with previous single-cell transcriptomic studies, we tried various strategies to confirm the identity of the two subpopulations of granule cells (GCs) derived from neuroblasts but didn’t get a clear answer. As a result, we can only provide an objective description of the differences in gene expression and developmental trajectory and speculate that these differences may be related to their degree of maturity but are not aligned on the same trajectory.

In the end, we have discussed the technical and conceptual limitations of this work and added a brief discussion about these limitations in the last paragraph of the main text. We hope the readers can interprate our data critically and objectively.

**Reviewer #2 (Public Review):**
In this manuscript, Yao et al. present a series of experiments aiming at generating a cellular atlas of the human hippocampus across aging, and how it may be affected by injury, in particular, stroke. Although the aim of the study is interesting and relevant for a larger audience, due to the ongoing controversy around the existence of adult hippocampal neurogenesis in humans, a number or technical weaknesses result in a poor support for many of the conclusions made from the results of these experiments.In particular, a recent meta analysis of five previous studies applying similar techniques to human samples has identified different aspects of sample size as main determinants of the statistical power needed to make significant conclusions. Some of this aspects are the number of nuclei sequenced and subject stratification. These two aspects are of concern in Yao's study. First, the number of sequenced nuclei is lower than the calculated numbers of nuclei required for detecting rare cell types. However, Yao et al. report succeeding in detecting rare populations, including several types of neural stem cells in different proliferation states, which have been demonstrated to be extremely scarce by previous studies. It would be very interesting to read how the authors interpret these differences. Secondly, the number of donors included in some of the groups is extremely low (n=1) and the miscellaneous information provided about the donors is practically inexistent. As individual factors such as chronic conditions, medication, lifestyle parameters, etc... are considered determinant for the variability of adult hippocampal neurogenesis levels across individuals, this represents a series limitation of the current study. Overall, several technical weaknesses severely limit the relevance of this study and the ability of the authors to achieve their experimental aims.After a first review round, the manuscript is still lacking a clear discussion of its several technical limitations, which will help the audience to grasp the relevance of the findings. In particular, detailed information about individual patients health status and relevant lifestyle parameters that may have affected it is lacking. The authors make the point themselves that the discrepancies among studies might be caused by health state differences across hippocampi, which subsequently lead to different degrees of hippocampal neurogenesis.". So, even in the authors own interpretation this is a serious limitation to the manuscript, that however out of the authors control, impacts on the quality of their findings.
**Reviewer #2 (Recommendations For The Authors):**
Please see public review. I do understand the authors point about incomplete patient data collection and low patient numbers and how the former is out of their control. Nevertheless, these are crucial parameters that impact negatively on the quality and relevance of several of their bold claims in the manuscript, especially given the low number of patients included. The current version still lacks a clear and honest discussion of the several technical and conceptual limitations of the authors work, as in some cases they are presented to the reviewers in the rebuttal letter, for the readership, so that they could critically evaluate the relevance of the authors' finding in a bigger perspective.

We thank the reviewer for reevaluating our revised manuscript. We respect the reviewer’s comm¬ents and discuss the technical and conceptual limitations of this work. Here we provide the response to Reviewer #2 (Public Review) on these below.

We understand the reviewer’s concern and have also noticed that according to the computational modeling conducted by Tosoni et al. (Neuron, 2023), at least 21 neuroblast cells (NBs) can be identified out of 30,000 granule cells (GCs) from a total of 180,000 dentate gyrus (DG) cells. In our dataset, we sequenced 24,671 GC nuclei and 92,966 total DG cell nuclei, which also includes neonatal samples. The number of nuclei we sequenced is 4.5 times higher than that of Wang et al. (Cell Research, 2022), who also detected NBs. Therefore, it is possible that we are able to detect NBs. Importantly, we have implemented strict quality control measures to support the reliability of our sequencing data. These measures include: 1. Immediate collection of tissue samples after postmortem (3-4 hrs) to ensure the quality of isolated nuclei. 2. Only nuclei expressing more than 200 genes but fewer than 5000-8600 genes (depending on the peak of enrichment genes) were considered. On average, each cell detected around 3000 genes. 3. The average proportion of mitochondrial genes in each sample was approximately 1.8%, with no sample exceeding 5%. We have shown that the number of cells captured from individual samples and the average number of genes detected per cell are sufficient, indicating overall good sequencing quality (Figure 1-supplement 1A,B andF, and Figure 1-source data 1). Additionally, we have further confirmed the presence of these cell types with low abundance by integrating immunofluorescence staining (Figure 4E, 5D and 6B), cell type-specific gene expression (Figure1 C and D), overall transcriptomic characteristics (Figure 1-supplement 1E), and developmental potential (Figure4 A-D, Figure 6E and F). We hope these evidences together could explain why we can identify the rare neurogenic populations.

Regarding the limited sample size and poor patient information, we cannot adequately address these two major concerns. Due to the scarcity of stroke or neonatal human samples, it was not feasible to collect a larger sample size within the expected timeframe. We have collected additional details about the donors. Please refer to Figure 1-source data 1 for the updated information. Other information regarding the lifestyle parameters of these donors has not been sufficiently recorded by the hospital. Therefore, we cannot improve the patient information further.

As per the reviewer’s recommendation, in the latest version, we have discussed the technical and conceptual limitations of this work and added a brief discussion about these limitations in the last paragraph of the main text. We hope the readers can interprate our data critically and objectively.